# STIM1-dependent peripheral coupling governs the contractility of vascular smooth muscle cells

**Vivek Krishnan[1†], Sher Ali[1†], Albert L Gonzales[2†], Pratish Thakore[1†], Caoimhin S Griffin[1], Evan Yamasaki[1], Michael G Alvarado[1], Martin T Johnson[3], Mohamed Trebak[3,4], Scott Earley[1]\***

[1]Department of Pharmacology, Center for Molecular and Cellular Signaling in the Cardiovascular System, University of Nevada, Reno, United States; [2]Department of Physiology and Cell Biology, Center for Molecular and Cellular Signaling 18 in the Cardiovascular System, University of Nevada, Reno, United States; [3]Department of Cellular and Molecular Physiology, Penn State Cancer Institute, Penn State University, Reno, United States; [4]Department of Pharmacology and Chemical Biology, and Vascular Medicine Institute, University of Pittsburgh, Pittsburgh, United States

**Abstract** Peripheral coupling between the sarcoplasmic reticulum (SR) and plasma membrane (PM) forms signaling complexes that regulate the membrane potential and contractility of vascular smooth muscle cells (VSMCs). The mechanisms responsible for these membrane interactions are poorly understood. In many cells, STIM1 (stromal interaction molecule 1), a single-transmembrane-domain protein that resides in the endoplasmic reticulum (ER), transiently moves to ER-PM junctions in response to depletion of ER $Ca^{2+}$ stores and initiates store-operated $Ca^{2+}$ entry (SOCE). Fully differentiated VSMCs express STIM1 but exhibit only marginal SOCE activity. We hypothesized that STIM1 is constitutively active in contractile VSMCs and maintains peripheral coupling. In support of this concept, we found that the number and size of SR-PM interacting sites were decreased, and SR-dependent $Ca^{2+}$-signaling processes were disrupted in freshly isolated cerebral artery SMCs from tamoxifen-inducible, SMC-specific STIM1-knockout (*Stim1*-smKO) mice. VSMCs from *Stim1*-smKO mice also exhibited a reduction in nanoscale colocalization between $Ca^{2+}$-release sites on the SR and $Ca^{2+}$-activated ion channels on the PM, accompanied by diminished channel activity. *Stim1*-smKO mice were hypotensive, and resistance arteries isolated from them displayed blunted contractility. These data suggest that STIM1 – independent of SR $Ca^{2+}$ store depletion – is critically important for stable peripheral coupling in contractile VSMCs.

**\*For correspondence:**
searley@med.unr.edu

[†]These authors contributed equally to this work

## Introduction

Subcellular $Ca^{2+}$-signaling microdomains formed by interactions between the sarcoplasmic reticulum (SR) and the plasma membrane (PM) are vital for many physiological processes, including regulation of the contractility of vascular smooth muscle cells (VSMCs) (*Nelson et al., 1995*; *Gonzales et al., 2014*). $Ca^{2+}$ signals that occupy these compartments are typified by $Ca^{2+}$ sparks – large-amplitude $Ca^{2+}$ transients that reflect optically detected $Ca^{2+}$ ions released into the cytosol from the SR through clusters of type 2 ryanodine receptors (RyR2s). $Ca^{2+}$ sparks activate clusters of large-conductance $Ca^{2+}$-activated $K^+$ (BK) channels on the PM, generating transient, macroscopic outward $K^+$ currents that hyperpolarize the PM (*Nelson et al., 1995*; *Mironneau et al., 1996*; *ZhuGe et al., 1999*). A complementary $Ca^{2+}$-signaling pathway that causes VSMC membrane depolarization and elevated contractility is formed by interactions between inositol 1,4,5-trisphosphate receptors (IP₃Rs) on the

SR and monovalent cation-selective, $Ca^{2+}$-activated TRPM4 (transient receptor potential melastatin 4) channels on the PM. $Ca^{2+}$ released from the SR through $IP_3Rs$ activates $Na^+$ influx through TRPM4, causing depolarization of the PM and increased VSMC contractility (*Gonzales et al., 2014*; *Gonzales et al., 2010b*). The close association of the SR and PM creates subcellular compartments where the local $Ca^{2+}$ ion concentration can reach the micromolar range required for activation of BK and TRPM4 channels under physiological conditions (*Zhuge et al., 2002*). In nonexcitable cells, endoplasmic reticulum (ER)-PM junctions and associated proteins have been well characterized (*Chang et al., 2017*; *Chen et al., 2019*). In contrast, SR-PM junctional areas of VSMCs and the essential proteins that mediate these interactions remain poorly understood.

The ER-PM junctions of nonexcitable cells are highly specialized hubs for ion channel signaling cascades. These spaces are the sites of one of the most ubiquitous receptor-regulated $Ca^{2+}$ entry pathways in such cells, termed store-operated $Ca^{2+}$ entry (SOCE), which is mediated by the ER-resident $Ca^{2+}$-sensing protein STIM1 (stromal interaction molecule 1) and $Ca^{2+}$-selective channels of the Orai group on the PM (*Michaelis et al., 2015*; *Mercer et al., 2006*; *Kwon et al., 2017*; *Prakriya and Lewis, 2015*; *Emrich et al., 2022*). STIM1 is a single-pass transmembrane ER/SR protein that possesses a low-affinity $Ca^{2+}$-sensing EF-hand facing the lumen of the ER/SR (*Michaelis et al., 2015*; *Abdullaev et al., 2008*; *Berry et al., 2011*; *Chin-Smith et al., 2014*; *Correll et al., 2015*; *Jones et al., 2008*; *Klejman et al., 2009*; *Koh et al., 2009*; *López et al., 2008*; *Lu et al., 2010*; *Lu et al., 2008*; *Lyfenko and Dirksen, 2008*; *Numaga-Tomita and Putney, 2013*; *Nurbaeva et al., 2015*; *Onodera et al., 2013*; *Peel et al., 2006*; *Takahashi et al., 2007*; *Wissenbach et al., 2007*; *Zhang et al., 2007*; *Zhou et al., 2015*). Following $Ca^{2+}$ store depletion by $IP_3$-producing receptor agonists, STIM1 acquires an extended conformation and migrates to ER-PM junctions, exposing a cytosolic STIM-Orai-activating region that physically traps and activates Orai channels on the PM (*Michaelis et al., 2015*; *Mercer et al., 2006*; *Abdullaev et al., 2008*; *Koh et al., 2009*; *Lyfenko and Dirksen, 2008*; *Nurbaeva et al., 2015*; *Peel et al., 2006*; *Zhang et al., 2007*; *Zhou et al., 2015*; *Perni et al., 2015*; *Soboloff et al., 2006*; *Spassova et al., 2006*; *Stathopulos et al., 2013*; *Zhou et al., 2013*). The other STIM protein family member, STIM2, is structurally similar to STIM1. Fully differentiated VSMCs from systemic arteries express STIM1 but not STIM2 and do not exhibit detectable SOCE or its biophysical manifestation, the $Ca^{2+}$ release-activated $Ca^{2+}$ (CRAC) current (*Bisaillon et al., 2010*; *Potier et al., 2009*; *Fernandez et al., 2015*). Many species express $IP_3Rs$ but lack STIM and Orai proteins, suggesting that receptor-evoked $Ca^{2+}$ signaling is not always complemented by the operation of STIM and Orai mechanisms (*Collins and Meyer, 2011*). Evolutionary evidence indicates that Orai appeared before STIM, implying that STIM might have arisen to support the function of ER-PM junctions and only subsequently co-opted an existing Orai for SOCE (*Collins and Meyer, 2011*). Additional accumulating evidence indicates that, in addition to its role in SOCE, mammalian STIM1 protein serves as an essential regulator of several other ion channels and signaling pathways. STIM1 both positively and negatively regulates the function of L-type voltage-gated $Ca^{2+}$ channels (Cav1.2) (*Harraz and Altier, 2014*), transient receptor potential canonical (TRPC) channels (*Worley et al., 2007*), and arachidonate-regulated $Ca^{2+}$ (ARC) channels (*Mignen et al., 2007*). It has also been reported to regulate the function of $Ca^{2+}$ pumps, such as the SR/ER $Ca^{2+}$ ATPase (SERCA) and PM $Ca^{2+}$ ATPase (PMCA), as well as several cAMP-producing adenylyl cyclases at the PM (*Lee et al., 2014*; *Ritchie et al., 2012*; *Martin et al., 2009*; *Motiani et al., 2018*).

In this study, we investigated the role of STIM1 in the formation of stable peripheral coupling sites in native, contractile SMCs from cerebral arteries. We show that STIM1 knockout disrupts the functional coupling of $Ca^{2+}$ release sites on the SR with $Ca^{2+}$-dependent ion channels on the PM. We further show that this function of STIM1 is independent of Orai1 channel activity and SR $Ca^{2+}$ store depletion and acts to sustain subcellular $Ca^{2+}$-signaling pathways that are essential for the regulation of VSMC contractility.

## Results

### *Stim1*-smKO mice lack STIM1 protein expression in VSMCs

Mice with *loxP* sites flanking exon 2 of the *Stim1* gene (*Stim1^{fl/fl}* mice) were crossed with myosin heavy chain 11 *Myh11^{CreERT2}* mice (*Chappell et al., 2016*; *Wirth et al., 2008*), generating *Myh11^{CreERT2}: Stim1^{fl/wt}* mice, in which *Myh11* promoter-driven *Cre* expression is induced by injection of tamoxifen.

Heterozygous *Myh11^CreERT2^: Stim1^fl/wt^* mice were then intercrossed, yielding tamoxifen-inducible *Myh11^CreERT2^: Stim1^fl/fl^* mice. Cre-recombinase expression was induced in male *Myh11^CreERT2^: Stim1^fl/fl^* mice by daily intraperitoneal injection of tamoxifen (100 µL, 10 mg/mL) for 5 days, beginning at 4–6 weeks of age to generate SMC-specific *Stim1* knockout mice (*Stim1*-smKO). Controls for all experiments consisted of *Myh11^CreERT2^: Stim1^fl/fl^* mice injected with sunflower oil, the vehicle for tamoxifen. Mice were used for experiments 1 week after the final injection. The Wes capillary electrophoresis immunoassay-based protein detection system was used for qualitative and quantitative assessment of STIM1 protein in smooth muscle tissues from *Stim1*-smKO and control mice. STIM1 protein was readily detected as a single band in cerebral artery, mesenteric artery, aortic, colonic, and bladder smooth muscle isolated from control mice but was virtually undetectable in smooth muscle isolated from *Stim1*-smKO mice (*Figure 1A*). STIM1 protein levels normalized to total protein (*Figure 1— figure supplement 1A*) were significantly lower in cerebral artery, aortic, colonic, and bladder smooth muscle from *Stim1*-smKO mice compared with controls (*Figure 1A*). In contrast, STIM1 protein expression was detected at similar levels in whole brains from both control and *Stim1*-smKO mice (*Figure 1A*), reflecting STIM1 expression in brain cells apart from VSMCs. Tamoxifen injection had no effect on STIM1 protein levels in *Myh11^CreERT2^: Stim1^wt/wt^* mice (*Figure 1—figure supplement 1B–G*).

In further studies, single SMCs from cerebral arteries isolated from control and *Stim1*-smKO mice were enzymatically dispersed, immunolabeled with an anti-STIM1 primary antibody, and imaged using a ground state depletion followed by individual molecule return (GSDIM) superresolution microscopy system in epifluorescence (*Figure 1B*) and total internal reflection fluorescence (TIRF) (*Figure 1— figure supplement 2A*) modes. We previously showed that our GSDIM system has a lateral resolution of 20–40 nm (*Pritchard et al., 2019*; *Thakore, 2020*). TIRF-mode GSDIM detects fluorophores at or near the PM to a depth of approximately 150 nm.

VSMCs from control mice exhibited punctate STIM1 protein clusters (*Figure 1B*). Frequency analyses revealed that the sizes of these clusters were exponentially distributed, with a majority of clusters (~95%) ranging in area between 400 and 7600 $nm^2$ (mean = 2135 ± 21 $nm^2$; median = 800 $nm^2$) (*Figure 1C*). STIM1 cluster density and size were significantly reduced in VSMCs isolated from *Stim1*-smKO mice (*Figure 1D*, *Figure 1—figure supplement 2B and C*). In addition, the number of GSDIM events detected in VSMCs isolated from *Stim1*-smKO mice was comparable to background levels observed in cells from control mice immunolabeled with secondary antibody only, providing further evidence of effective STIM1 knockdown (*Figure 1—figure supplement 2D and E*). Taken together, these data demonstrate selective, tamoxifen-inducible SMC-specific knockout of STIM1 expression in *Stim1*-smKO mice.

In many cells, depletion of ER/SR $Ca^{2+}$ stores causes STIM1 to form large clusters and initiate SOCE. Here, we examined the effects of the SERCA pump inhibitor thapsigargin on STIM1 clusters in VSMCs from control mice using TIRF-mode GSDIM. This treatment had no effect on the density or size of STIM1 protein clusters at the PM (*Figure 1—figure supplement 3A–C*). In addition, we compared SOCE between cultured, proliferative cerebral artery VSMCs and native, contractile VSMCs. We found that proliferative VSMCs exhibit robust SOCE, whereas SOCE is virtually undetectable in contractile VSMCs (*Figure 1—figure supplement 3D and E*). These data indicate that in contractile VSMCs STIM1 cluster size and density are unaffected by the depletion of SR $Ca^{2+}$ and that SOCE is absent from these cells.

## PM and SR coupling is diminished in VSMCs from *Stim1*-smKO mice

To investigate how STIM1 knockout affects PM and SR interactions, we costained native SMCs isolated from cerebral arteries of control and *Stim1*-smKO mice with Cell-Mask Deep Red and ER-Tracker Green to label the PM and SR, respectively, as described in our prior publications (*Pritchard et al., 2019*; *Pritchard et al., 2017*). Using live-cell structured illumination microscopy (SIM), we acquired Z-stack images of PM- and SR-labeled VSMCs as 0.25 µm slices. We then reconstructed the 3D surfaces of the PM and SR from these images (*Figure 2A*), also generating a third surface indicating the sites of colocalization between the PM and SR (*Figure 2A*, *Videos 1 and 2*). The mean volume of the PM did not differ between *Stim1*-smKO and control mice, but the volume of the SR was smaller in cells isolated from *Stim1*-smKO mice (*Figure 2B and C*). The reduction in SR volume is likely due to the peripheral SR pulling away from the PM. The overall PM-SR colocalization was significantly reduced in VSMCs from *Stim1*-smKO mice compared with controls (*Figure 2D*). As shown in representative

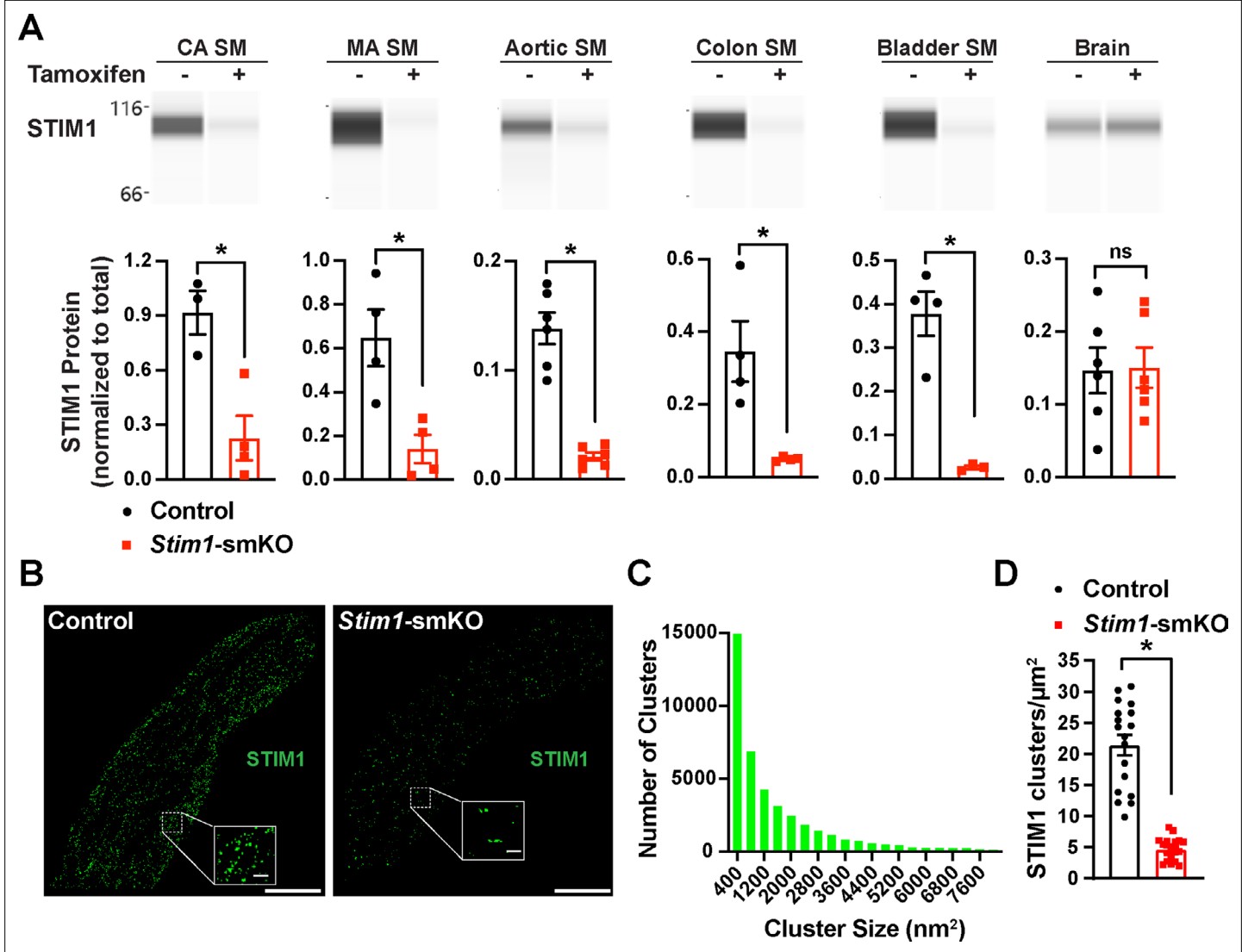

**Figure 1.** Inducible smooth muscle cell (SMC)-specific *Stim1* knockout. (**A**) Representative Wes protein capillary electrophoresis experiments, presented as Western blots, showing STIM1 protein expression levels in smooth muscle tissues and brains of control and *Stim1*-smKO mice. Summary data showing densitometric analyses of STIM1 protein expression in cerebral artery smooth muscle (CA SM), mesenteric artery smooth muscle (MA SM), aortic smooth muscle, colonic smooth muscle, bladder smooth muscle, and brain, normalized to total protein (n = 3–6 mice/group; *p<0.05, unpaired *t*-test). ns, not significant. (**B**) Representative epifluorescence superresolution localization maps of isolated cerebral artery SMCs from control and *Stim1*-smKO mice immunolabeled for STIM1. Insets: enlarged areas highlighted by the white squares in the main panels. Scale bars: 3 µm (main panels) and 250 nm (inset panels). (**C**) Distribution plot of the surface areas of individual STIM1 clusters in cerebral artery SMCs isolated from control mice (n = 42,726 clusters from 18 cells from three mice). (**D**) STIM1 cluster density in cerebral artery SMCs isolated from control and *Stim1*-smKO mice (n = 18 cells from three mice/group; *p<0.05, unpaired *t*-test).

The online version of this article includes the following source data and figure supplement(s) for figure 1:

**Source data 1.** Individual data points and analysis summaries for datasets shown in *Figure 1*.

**Figure supplement 1.** STIM1 protein expression remains unaltered in tamoxifen-injected *Myh11^CreERT2^: Stim1^fl/fl^* mice.

**Figure supplement 1—source data 1.** Individual data points and analysis summaries for datasets shown in *Figure 1—figure supplement 1*.

**Figure supplement 2.** Cerebral artery SMCs from *Stim1*-smKO mice exhibit reduced STIM1 cluster size and density.

**Figure supplement 2—source data 1.** Individual data points and analysis summaries for datasets shown in *Figure 1—figure supplement 2*.

**Figure supplement 3.** SR calcium store depletion does not affect STIM1 cluster size or density in isolated cerebral artery SMCs.

**Figure supplement 3—source data 1.** Individual data points and analysis summaries for datasets shown in *Figure 1—figure supplement 3*.

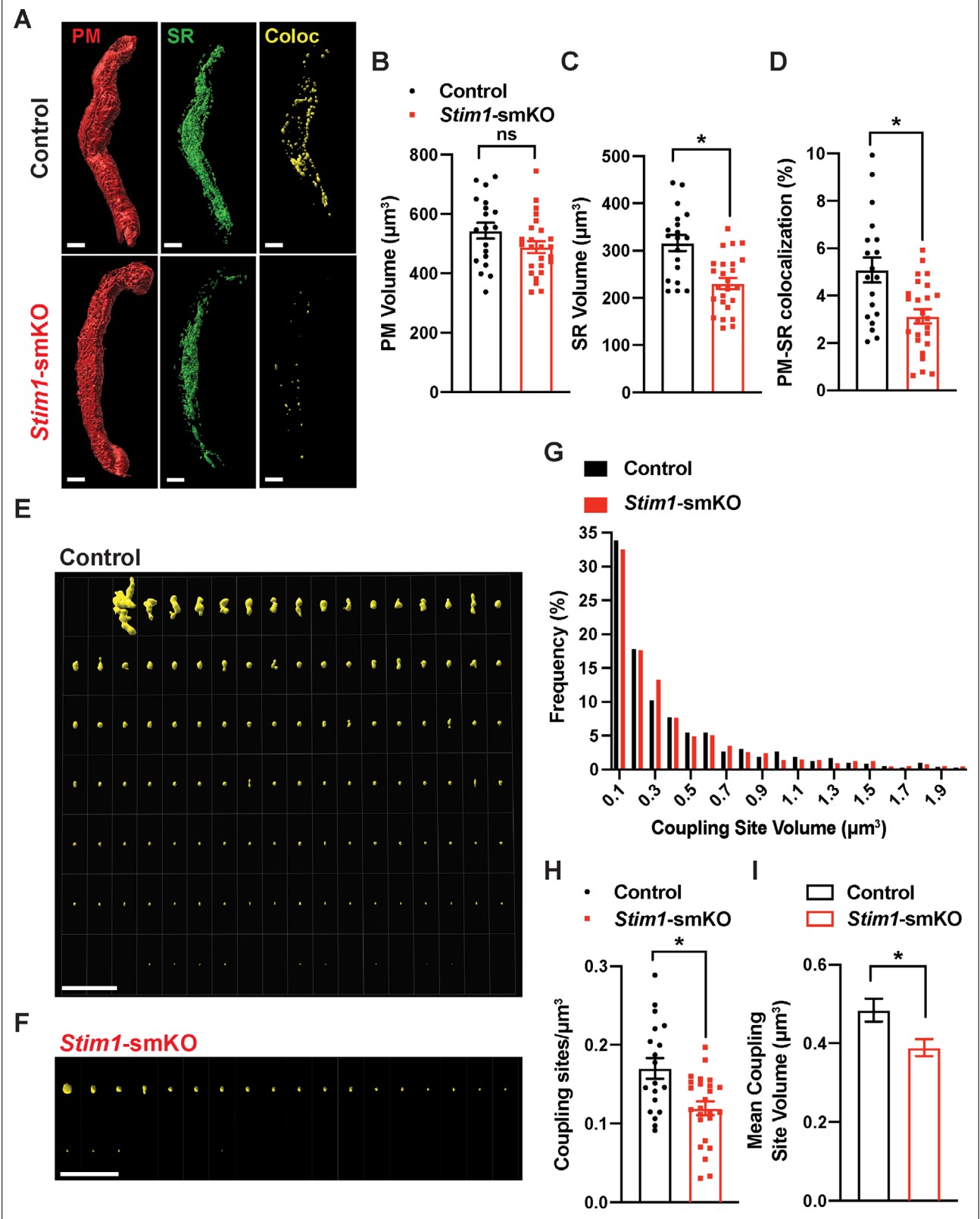

**Figure 2.** *Stim1* knockout decreases the density and area of plasma membrane-sarcoplasmic reticulum (PM-SR) coupling sites. (**A**) Representative 3D surface reconstructions of cerebral artery smooth muscle cells (SMCs) isolated from control and *Stim1*-smKO mice labeled with PM (red) and SR (green) dyes and imaged using structured illumination microscopy (SIM). Representations of colocalizing PM and SR surfaces (yellow), generated from surface reconstructions. Scale bar: 5 μm. (**B, C**) PM and SR volumes and (**D**) PM-SR colocalization (%) in cells from control and *Stim1*-smKO mice. (**E, F**) Ensemble

*Figure 2 continued on next page*

*Figure 2 continued*

images of all PM-SR colocalization sites in single cells from the control and *Stim1*-smKO mice shown in panel (**A**). Scale bar: 10 μm. (**G**) Frequency distribution of the volumes of individual PM-SR colocalization sites in VSMCs isolated from control and *Stim1*-smKO mice. (**H**) Densities and (**I**) mean volumes of individual coupling sites in VSMCs from control and *Stim1*-smKO mice. Data are for 1736 colocalization sites in 19 cells from six mice for control and 1484 colocalization sites in 25 cells from seven mice for *Stim1*-smKO (*p<0.05, unpaired *t*-test). ns, not significant.

The online version of this article includes the following source data for figure 2:

**Source data 1.** Individual data points and analysis summaries for datasets shown in *Figure 2*.

image galleries of individual colocalization sites (*Figure 2E and F*), the majority of PM-SR coupling sites in cells from both groups formed spherical surfaces, but some of the larger structures exhibited an elongated morphology. Frequency analyses showed that the volume of individual colocalization sites in cells from both groups exhibited an exponential distribution (*Figure 2G*). In addition, the number of coupling sites per unit volume and mean volume of individual sites were smaller in cells from *Stim1*-smKO mice compared with those from controls (*Figure 2H and I*). These data indicate that STIM1 maintains contact between the peripheral SR and PM, and interactions between the PM and SR are decreased by *Stim1* knockout in VSMCs.

### *Stim1* knockout decreases the colocalization of BK and RyR2 protein clusters

BK channels on the PM of VSMCs are functionally coupled with RyR2s on the SR (*Nelson et al., 1995*). Therefore, we investigated how *Stim1* knockout affects the nanoscale structure of BK-RyR2 signaling complexes using GSDIM superresolution microscopy. Freshly isolated VSMCs from *Stim1*-smKO and control mice were co-immunolabeled for RyR2 and the BK channel pore-forming subunit BKα and imaged using GSDIM in epifluorescence illumination mode. The resulting superresolution localization maps (*Figure 3A*, leftmost panels) showed that both proteins were present as defined clusters in VSMCs. Using an object-based analysis (OBA) approach (*Bolte and Cordelières, 2006*; *Lachmanovich et al., 2003*) as described in previous publications (*Pritchard et al., 2019*; *Thakore, 2020*; *Pritchard et al., 2017*; *Griffin et al., 2020*; *Pritchard et al., 2018*), we generated new maps of RyR2 clusters that overlapped at the resolution limit of our microscope system (~20–40 nm) with the centroid of each BK cluster and BK clusters that coincided with the centroid of each RyR2 cluster (*Supplementary file 1*). These two maps were then merged to reveal colocalized RyR2-BK channel protein clusters in VSMCs from both groups of animals that were below the resolution of our GSDIM system (*Figure 3A*, middle and rightmost panels). Particle analysis of these clusters showed that the density of individual BK protein clusters (number of clusters per unit area)

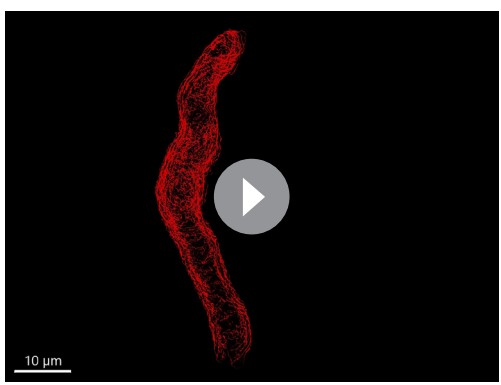

**Video 1.** Plasma membrane-sarcoplasmic reticulum (PM-SR) interactions in a cerebral artery smooth muscle cell (SMC) isolated from a control mouse. Animated representation of a SIM image series reconstructed and rendered in 3D. The PM is shown in red and made transparent for better visualization; the SR is shown in green, and colocalized areas are shown in yellow.
https://elifesciences.org/articles/70278/figures#video1

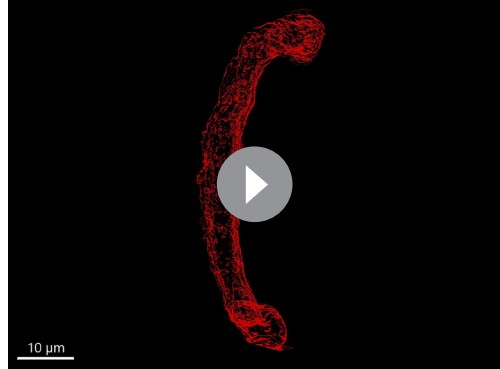

**Video 2.** Plasma membrane-sarcoplasmic reticulum (PM-SR) interactions in a cerebral artery smooth muscle cell (SMC) isolated from a *Stim1*-smKO mouse. Animated representation of a SIM image series reconstructed and rendered in 3D. The PM is shown in red and made transparent for better visualization; the SR is shown in green, and areas of colocalization are shown in yellow.
https://elifesciences.org/articles/70278/figures#video2

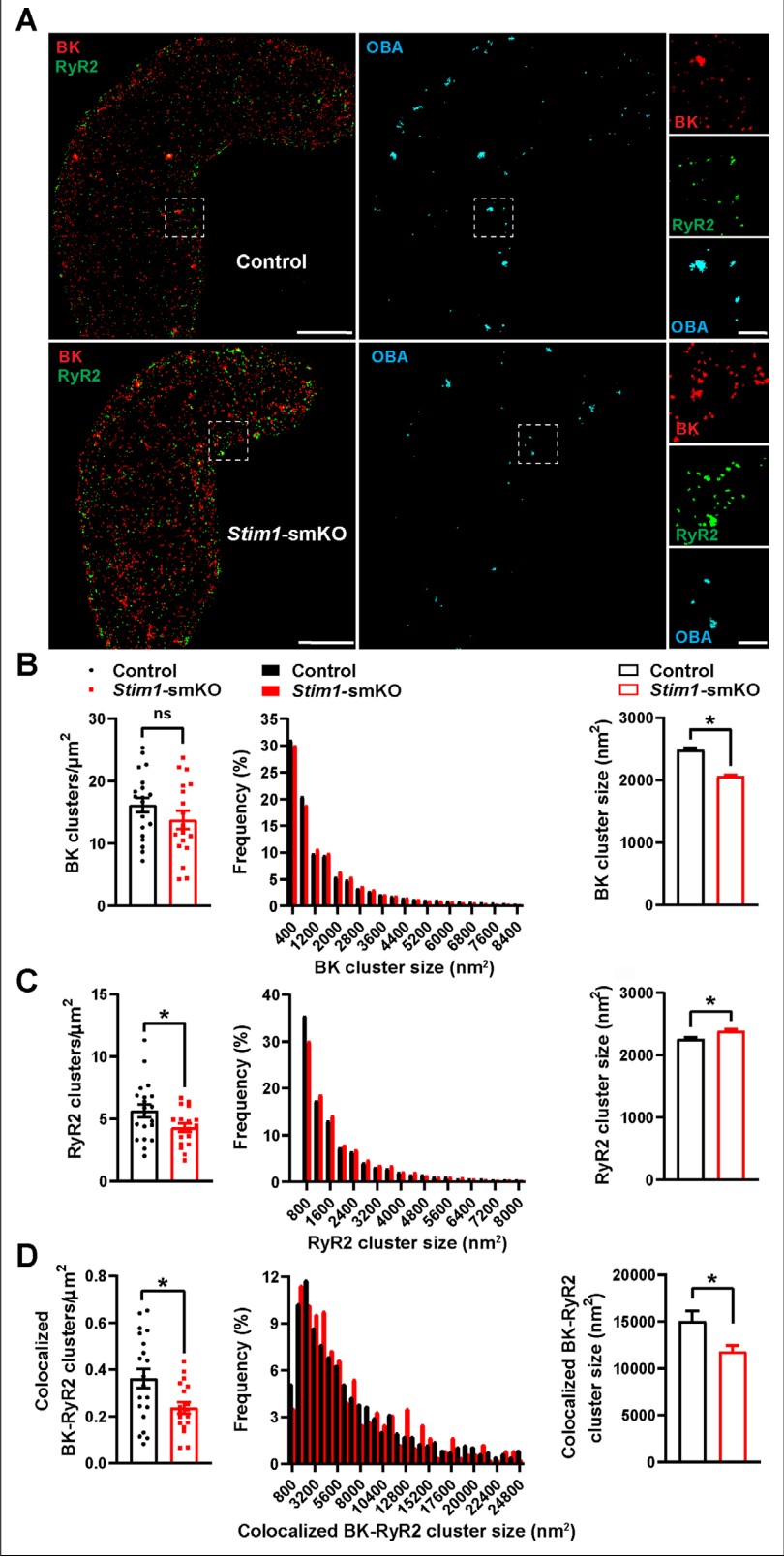

**Figure 3.** *Stim1* knockout decreases colocalization of BK and RyR2 protein clusters. (**A**) Epifluorescence-mode superresolution localization maps of freshly isolated vascular smooth muscle cells (VSMCs) from control and *Stim1*-smKO mice immunolabeled for BK (red) and RyR2 (green). Colocalized BK and RyR2 clusters were identified by object-based analysis (OBA) and mapped (cyan). Scale bar: 3 μm. Panels to the right show enlarged areas of the

*Figure 3 continued on next page*

*Figure 3 continued*

original superresolution maps indicated by the white boxes. Scale bar: 500 nm. (**B**) Summary data showing the density (clusters per unit area), frequency distribution of sizes, and mean size of BK channel clusters. (**C**) Summary data showing the density, frequency distribution of sizes, and mean size of RyR2 clusters. (**D**) Summary data showing the density, frequency distribution of sizes, and mean size of colocalizing BK and RyR2 clusters, identified using OBA. For density data, n = 20 cells from three mice for controls and n = 18 cells from three mice for *Stim1*-smKO mice. For frequency distribution and mean cluster size data: control, n = 44,340 BK channel clusters, n = 15,193 RyR2 clusters, and n = 1054 colocalizing clusters; Stim1-smKO: n = 30,552 BK channel clusters, n = 9702 RyR2 clusters, and n = 547 colocalizing clusters (*p<0.05, unpaired *t*-test). ns, not significant.

The online version of this article includes the following source data for figure 3:

**Source data 1.** Individual data points and analysis summaries for datasets shown in *Figure 3*.

---

was similar for both groups of animals (*Figure 3B*), whereas the density of individual RyR2 clusters was lower in VSMCs from *Stim1*-smKO mice compared with controls (*Figure 3C*). In both groups, the sizes of individual BK channel and RyR2 clusters followed an exponential distribution (*Figure 3B and C*). The mean size of individual BK clusters was smaller in VSMCs from *Stim1*-smKO mice compared with those from controls (*Figure 3B*); in contrast, the mean size of RyR2 clusters was slightly larger in cells from *Stim1*-smKO mice (*Figure 3C*). In terms of colocalization, this analysis showed a significant reduction in the density of colocalized BK-RyR2 protein clusters in VSMCs from *Stim1*-smKO mice compared with controls (*Figure 3D*). The mean size of colocalizing clusters from *Stim1*-smKO mice was smaller compared with those from control mice (*Figure 3D*), and the sizes of BK-RyR2 colocalization sites in cerebral artery SMCs from both groups exhibited an exponential distribution (*Figure 3D*).

## *Stim1* knockout decreases the colocalization of TRPM4 and IP$_3$R protein clusters

TRPM4 channels on the PM are functionally coupled with IP$_3$Rs on the SR (*Gonzales et al., 2014*). Therefore, we also investigated how interactions between PM TRPM4 channels and SR IP$_3$Rs were altered by *Stim1* knockout. Freshly isolated VSMCs from control and *Stim1*-smKO mice were co-immunolabeled for TRPM4 and IP$_3$R and imaged using GSDIM in epifluorescence illumination mode. The resulting GSDIM localization maps showed that these proteins are present as discrete clusters in cells (*Figure 4A*, leftmost panels).

We next used OBA to identify and map individual and colocalized TRPM4 and IP$_3$R protein clusters (*Figure 4A*, middle and rightmost panels; *Supplementary file 1*). This analysis showed that the densities of individual TRPM4 and IP$_3$R clusters were similar in both groups (*Figure 4B and C*) and that their sizes were exponentially distributed (*Figure 4B and C*). The mean sizes of individual TRPM4 and IP$_3$R clusters were smaller in VSMCs from *Stim1*-smKO mice compared with those from controls (*Figure 4B and C*). The density of colocalized TRPM4-IP$_3$R cluster sites did not differ between groups (*Figure 4D*), but the sizes of these colocalized clusters were smaller in cells from *Stim1*-smKO mice compared with those from controls (*Figure 4D*). Like individual clusters, colocalized clusters exhibited an exponential distribution (*Figure 4D*).

## STIM1 colocalizes with BK and TRPM4 channels

To determine the location of STIM1 clusters relative to BK and TRPM4 channel clusters at the PM, we imaged VSMCs from control mice that had been coimmunolabeled for BK and STIM1 (*Figure 5A*) or TRPM4 and STIM1 (*Figure 5B*) using TIRF-GSDIM. The frequency of colocalization of BK and STIM1 clusters and TRPM4 and STIM1 clusters was determined using OBA and mapped (*Figure 5A and B*). For comparison, new maps were generated from each original superresolution map that replicated the density and cluster size distribution, but a random location was assigned to each protein cluster. We then performed OBA for the simulated random distribution and compared the colocalization frequency of the original maps with the colocalization frequency of their randomized counterparts. The fraction of colocalized clusters in the original maps was greater than its randomized counterpart for every cell (*Figure 5C and D*). These data show that BK and STIM1 (*Figure 5C*) and TRPM4 and STIM1 (*Figure 5D*) colocalized more frequently than predicted if the distribution of protein clusters was random, suggesting a mechanistic basis of interaction.

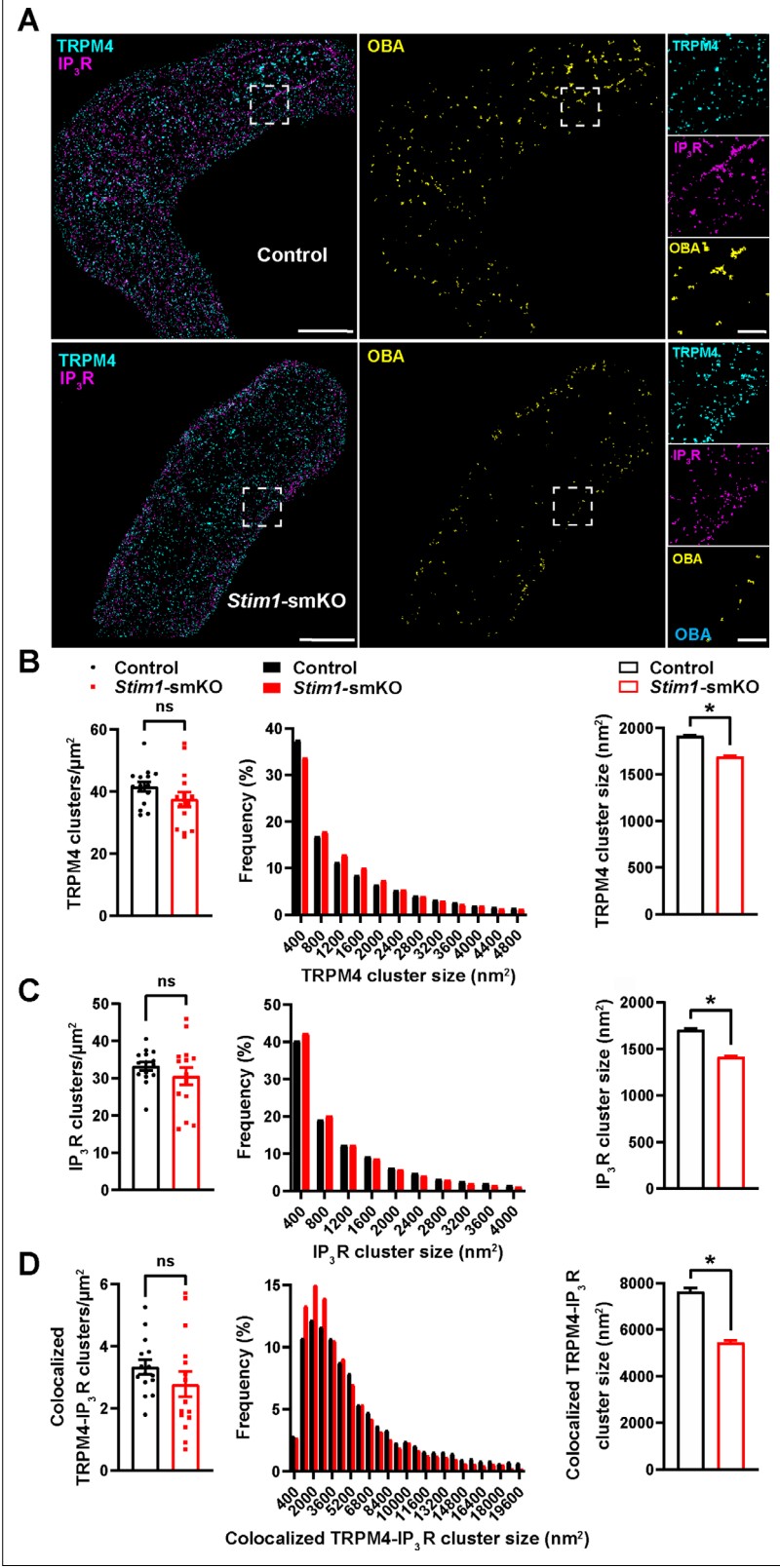

**Figure 4.** *Stim1* knockout decreases colocalization of TRPM4 and IP₃R protein clusters. (**A**) Epifluorescence-mode superresolution localization maps of freshly isolated vascular smooth muscle cells (VSMCs) from control and *Stim1*-smKO mice immunolabeled for TRPM4 (cyan) and IP₃R (magenta). Colocalized TRPM4 and IP₃R clusters were identified by object-based analysis (OBA) and mapped (yellow). Scale bar: 3 µm. Panels to the right show enlarged

*Figure 4 continued on next page*

*Figure 4 continued*

areas of the original superresolution maps indicated by white boxes. Scale bar: 500 nm. (**B**) Summary data showing the density (clusters per unit area), frequency distribution of sizes, and mean size of TRPM4 channel protein clusters. (**C**) Summary data showing the density, frequency distribution of sizes, and mean size of IP$_3$R clusters. (**D**) Summary data showing the density, frequency distribution of sizes, and mean size of colocalizing TRPM4 and IP$_3$R clusters, identified using OBA. For density data, n = 15 cells from three mice for both control and *Stim1*-smKO mice. For frequency distribution and mean cluster size data: control, n = 64,292 TRPM4 channel clusters, n = 51,728 IP$_3$R clusters, and n = 5164 colocalizing clusters; *Stim1*-smKO mice, n = 56,771 TRPM4 channel clusters, n = 45,717 IP$_3$R, and n = 3981 colocalizing clusters (*p<0.05, unpaired *t*-test). ns, not significant.

The online version of this article includes the following source data for figure 4:

**Source data 1.** Individual data points and analysis summaries for datasets shown in *Figure 4*.

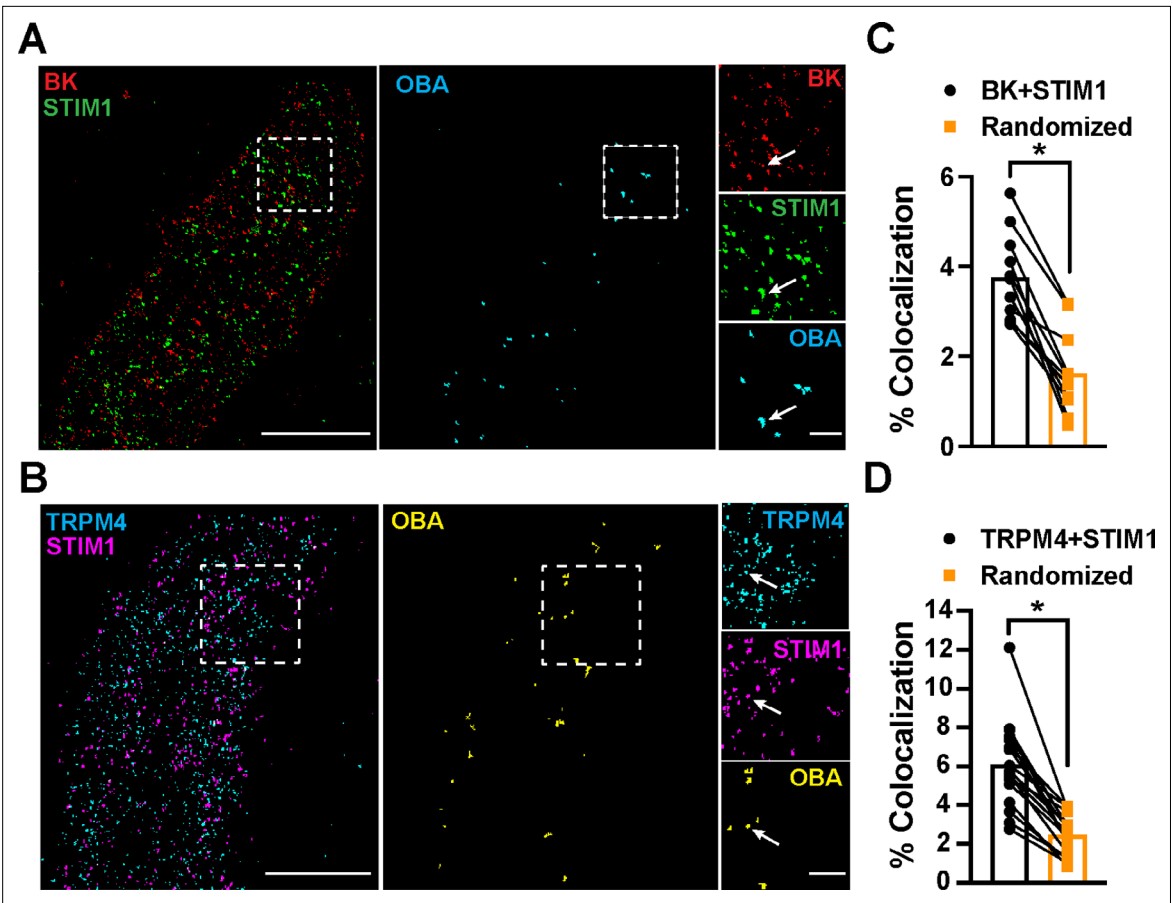

**Figure 5.** STIM1 colocalizes with BK and TRPM4. (**A**) Total internal reflection fluorescence (TIRF)-mode superresolution localization maps of freshly isolated vascular smooth muscle cells (VSMCs) from control mice immunolabeled for BK (red) and STIM1 (green). Colocalized BK and STIM1 clusters were identified by object-based analysis (OBA) and mapped (cyan). Scale bar: 3 μm. Panels to the right show enlarged areas of the original superresolution maps indicated by the white boxes. Arrows show examples of colocalizing clusters. Scale bar: 500 nm. (**B**) TIRF-mode superresolution localization maps of freshly isolated VSMCs from control mice immunolabeled for TRPM4 (cyan) and STIM1 (magenta). Colocalized TRPM4 and STIM1 clusters were identified by OBA and mapped (yellow). Scale bar: 2 μm. Panels to the right show enlarged areas of the original superresolution maps indicated by the white boxes. Arrows show examples of colocalizing clusters. Scale bar: 500 nm. (**C**) Colocalization frequency of BK and STIM1 clusters in imaged cells compared to colocalization frequency of BK and STIM1 clusters in randomized maps generated from respective cells. n = 11 cells from four mice (*p<0.05, paired *t*-test). (**D**) Colocalization frequency of TRPM4 and STIM1 clusters in imaged cells compared to colocalization frequency of TRPM4 and STIM1 clusters in randomized maps generated from respective imaged cells (n = 11 cells from four mice; *p<0.05, paired *t*-test).

The online version of this article includes the following source data for figure 5:

**Source data 1.** Individual data points and analysis summaries for datasets shown in *Figure 5*.

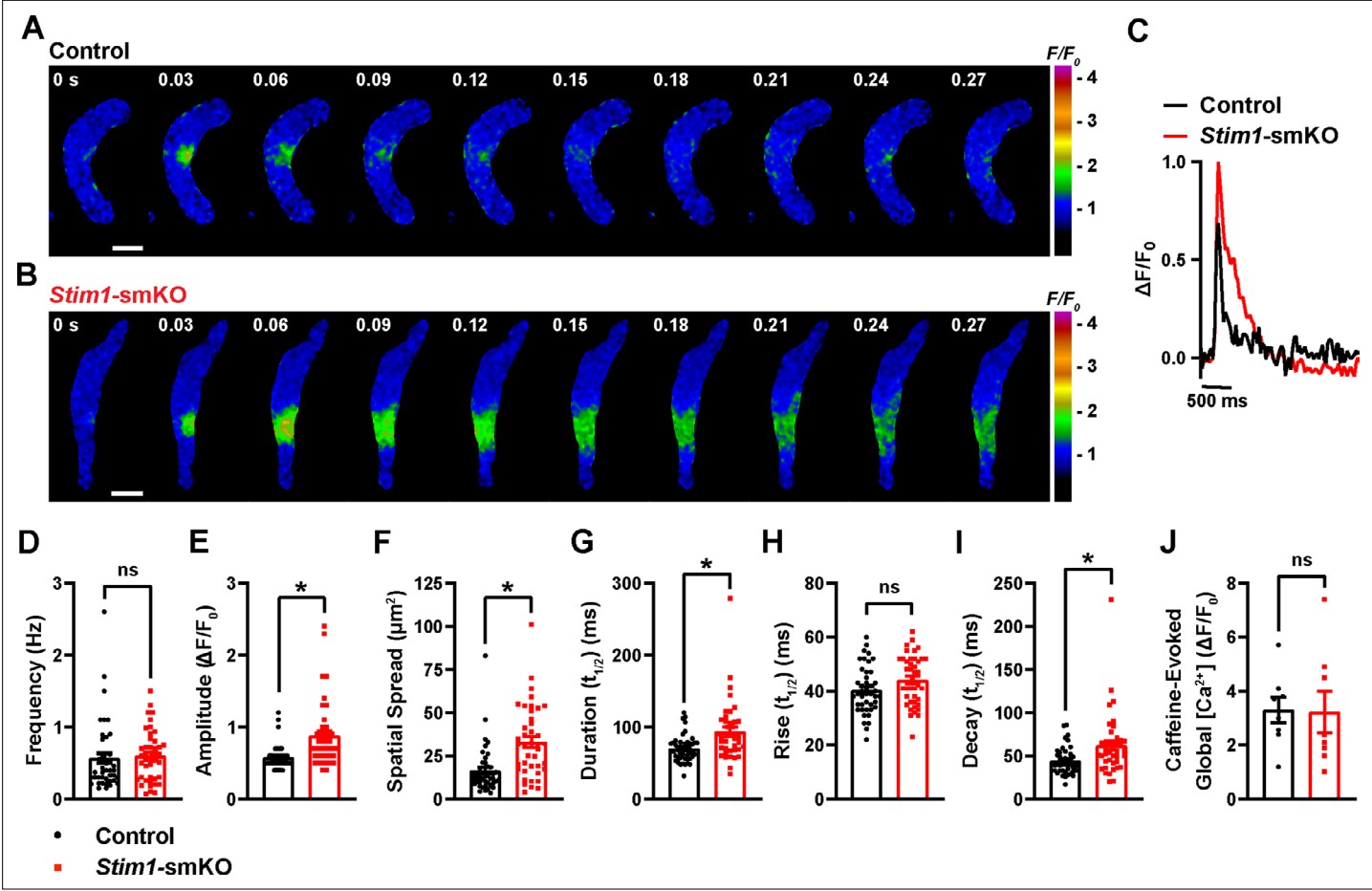

**Figure 6.** *Stim1* knockout alters Ca$^{2+}$ spark properties. (**A, B**) Representative time-course images of cerebral artery smooth muscle cells (SMCs) isolated from a control (**A**) or *Stim1*-smKO (**B**) mouse exhibiting Ca$^{2+}$ spark events, presented as changes in fractional fluorescence (F/F$_0$). The elapsed time of the event is shown in seconds (s). Scale bar: 10 μm. (**C**) Representative traces of Ca$^{2+}$ spark events in cerebral artery SMCs isolated from a control (black trace) or *Stim1*-smKO (red trace) mice presented as changes in fractional fluorescence (ΔF/F$_0$) vs. time. (**D–I**) Summary data showing Ca$^{2+}$ spark frequency (**D**), amplitude (**E**), spatial spread (**F**), event duration (**G**), rise time (**H**), and decay time (**I**) in vascular smooth muscle cells (VSMCs) isolated from control and *Stim1*-smKO mice (control, n = 43 spark sites in 18 cells from four mice; *Stim1*-smKO, n = 41 spark sites in 19 cells from four mice; *p<0.05, unpaired *t*-test). ns, not significant. (**J**) Summary data showing caffeine (10 mM)-evoked changes in global Ca$^{2+}$ in cerebral artery SMCs isolated from control and *Stim1*-smKO mice (control, n = 8 cells from four mice; *Stim1*-smKO, n = 8 cells from four mice, unpaired *t*-test). ns, not significant.

The online version of this article includes the following source data for figure 6:

**Source data 1.** Individual data points and analysis summaries for datasets shown in *Figure 6*.

## *Stim1* knockout alters the properties of Ca$^{2+}$ sparks

To investigate how *Stim1* knockout alters fundamental Ca$^{2+}$-signaling mechanisms, we loaded freshly isolated VSMCs with the Ca$^{2+}$-sensitive fluorophore Fluo-4 AM and imaged them using live-cell, high-speed, high-resolution spinning-disk confocal microscopy. Spontaneous Ca$^{2+}$ sparks were present in cerebral artery SMCs from both control (*Figure 6A*, *Video 3*) and *Stim1*-smKO (*Figure 6B*, *Video 4*) mice. The frequency of Ca$^{2+}$ spark events did not differ between groups (*Figure 6C and D*). However, the mean amplitude of Ca$^{2+}$ spark events was significantly greater in VSMCs isolated from *Stim1*-smKO mice compared with those from controls (*Figure 6E*). Further analyses revealed that spatial spreads, durations, and decay times of individual Ca$^{2+}$ spark events were significantly greater in VSMCs isolated from *Stim1*-smKO mice compared with those taken from control mice, but rise times did not differ (*Figure 6F–I*). To investigate the effects of *Stim1* knockout on total SR Ca$^{2+}$ store load, we applied a bolus of caffeine (10 mM) to Fluo-4 AM-loaded VSMCs isolated from control and *Stim1*-smKO mice. The peak amplitude of caffeine-evoked global increases in cytosolic [Ca$^{2+}$] did not differ between groups (*Figure 6J*), indicating that *Stim1* knockout did not alter total SR [Ca$^{2+}$]. Therefore,

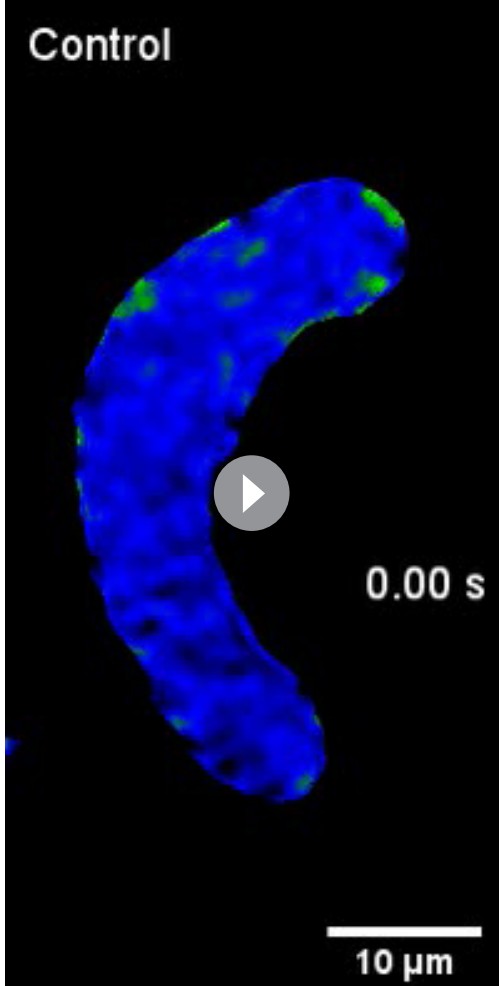

**Video 3.** Representative movie showing spontaneous Ca$^{2+}$ sparks in a cerebral artery smooth muscle cell (SMC) isolated from a control mouse.
https://elifesciences.org/articles/70278/figures#video3

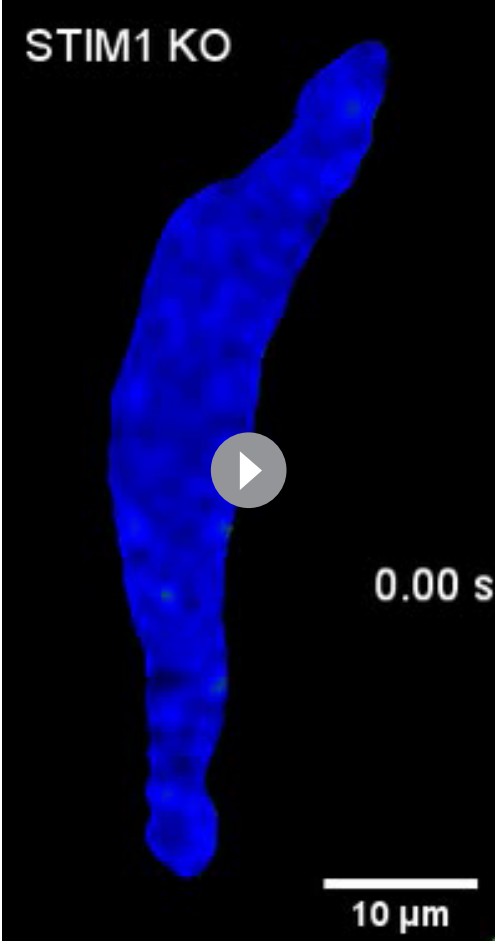

**Video 4.** Representative movie showing spontaneous Ca$^{2+}$ sparks in a cerebral artery smooth muscle cell (SMC) isolated from a *Stim1*-smKO mouse.
https://elifesciences.org/articles/70278/figures#video4

alterations in the properties of Ca$^{2+}$ sparks associated with the knockout of *Stim1* are not the result of changes in SR Ca$^{2+}$ load.

## *Stim1* knockout diminishes physiological BK and TRPM4 channel activity

We next used patch-clamp electrophysiology to investigate how knockout of *Stim1* affects the activity of BK and TRPM4 channels in VSMCs. When Ca$^{2+}$ sparks activate clusters of BK channels at the PM, they generate macroscopic K$^+$ currents termed spontaneous transient outward currents (STOCs) (*Nelson et al., 1995*). Here, we recorded STOCs over a range of membrane potentials using the amphotericin B perforated patch-clamp configuration, which allows the membrane potential to be controlled without disrupting intracellular Ca$^{2+}$-signaling pathways (*Pritchard et al., 2019*; *Pritchard et al., 2017*). The frequencies and amplitudes of STOCs were lower in VSMCs from *Stim1*-smKO mice compared with those from controls at all membrane potentials greater than –60 mV (*Figure 7A–C*). We measured whole-cell BK channel currents to determine if diminished STOC activity was attributable to a decrease in the total number of BK channels available for activation at the PM. Cerebral artery SMCs isolated from *Stim1*-smKO and control mice were patch-clamped in the conventional whole-cell configuration, and whole-cell K$^+$ currents were recorded during the application of voltage ramps. Using the selective BK blocker paxilline to isolate BK channel currents, we found that whole-cell BK current amplitude did not differ between VSMCs from control and *Stim1*-smKO mice (*Figure 7D and*

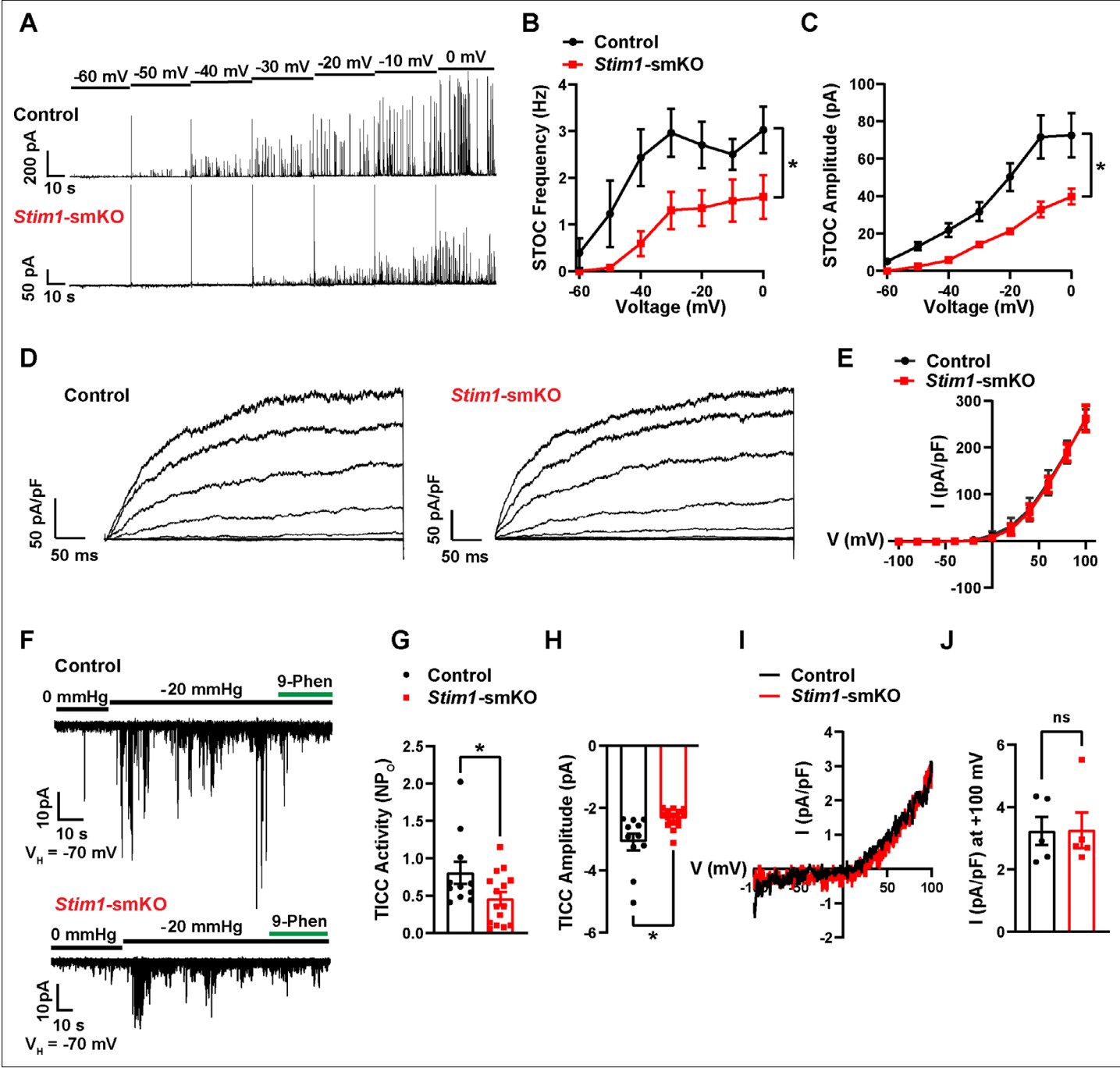

**Figure 7.** *Stim1* knockout diminishes physiological BK and TRPM4 channel activity. (**A**) Representative traces of spontaneous transient outward currents (STOCs) in cerebral artery smooth muscle cells (SMCs) from control and *Stim1*-smKO mice, recorded by perforated patch-clamp electrophysiology over a range of membrane potentials (−60 to 0 mV). (**B, C**) Summary data showing STOC frequency (**B**) and amplitude (**C**) (control, n = 13 cells from four animals; *Stim1*-smKO, n = 17 cells from five mice; *p<0.05, two-way ANOVA). (**D**) Representative traces of paxilline (1 μM)-sensitive BK currents in cerebral artery SMCs from control and *Stim1*-smKO mice, recorded by patch-clamping in conventional whole-cell mode during a series of command voltage steps (−100 to +100 mV). (**E**) Summary data for whole-cell BK currents (control, n = 6 cells from three mice; *Stim1*-smKO, n = 7 cells from three mice; two-way ANOVA). (**F**) Representative traces of TRPM4 currents in cerebral artery SMCs from control and *Stim1*-smKO mice voltage-clamped at –70 mV, recorded using perforated patch-clamp electrophysiology. TRPM4 currents were evoked as transient inward cation currents (TICCs) by application of negative pressure (–20 mmHg) through the patch pipette and were blocked by bath application of 9-phenanthrol (9-phen; 30 μM). (**G**) Summary data showing TICC activity as TRPM4 channel open probability ($NP_o$) and (**H**) TICC amplitude in control (n = 12 cells from five mice) and *Stim1*-smKO (n = 15 cells from five mice) mice (*p<0.05, unpaired *t*-test). (**I**) Representative conventional whole-cell patch-clamp recordings of 9-phenanthrol–sensitive TRPM4 currents in cerebral artery SMCs from control and *Stim1*-smKO mice. Currents were activated by free $Ca^{2+}$ (200 μM), included in the

*Figure 7 continued on next page*

*Figure 7 continued*

patch pipette solution, and were recorded using a ramp protocol from –100 to 100 mV from a holding potential of –60 mV. (**J**) Summary of whole-cell TRPM4 current density at +100 mV (control, n = 5 cells from three mice; *Stim1*-smKO, n = 5 cells from three mice, unpaired *t*-test). ns, not significant.

The online version of this article includes the following source data and figure supplement(s) for figure 7:

**Source data 1.** Individual data points and analysis summaries for datasets shown in *Figure 7*.

**Figure supplement 1.** Orai1 blockade by Synta66 does not affect STOC amplitude or frequency in cerebral artery SMCs.

**Figure supplement 1—source data 1.** Individual data points and analysis summaries for datasets shown in *Figure 7—figure supplement 1*.

**Figure supplement 2.** Orai1 blockade by Synta66 does not affect TICC activity in cerebral artery SMCs.

**Figure supplement 2—source data 1.** Individual data points and analysis summaries for datasets shown in *Figure 7—figure supplement 2*.

*E*), indicating that the number of BK channels available for activation and their functionality was not altered by *Stim1* knockout. *Stim1* knockout did not alter mRNA levels of BK α- or β1-subunits or RyR2s in cerebral arteries (*Figure 7—figure supplement 1A*). In addition, the potent and selective Orai1 blocker Synta66 (*Zhang et al., 2020*) had no effect on STOC amplitude or frequency (*Figure 7—figure supplement 1B–D*). These findings indicate that diminished STOC activity following knockout of *Stim1* is not due to changes in BK and RyR2 expression and that CRAC channel activity is not required for the generation of STOCs.

TRPM4 is a $Ca^{2+}$-activated, monovalent cation-selective channel that is impermeable to divalent cations (*Launay et al., 2002*). At membrane potentials in the physiological range for VSMCs (–70 to –30 mV), TRPM4 channels conduct inward $Na^+$ currents that depolarize the PM in response to increases in intraluminal pressure and receptor-dependent vasoconstrictor agonists (*Earley et al., 2007*; *Gonzales et al., 2010a*). Under native conditions, TRPM4 channels are activated by $Ca^{2+}$ released from the SR through $IP_3Rs$, generating transient inward cation currents (TICCs) (*Gonzales et al., 2014*). To determine the effects of STIM1 knockout on TRPM4 activity, we recorded TICCs using the amphotericin B perforated patch-clamp configuration (*Gonzales et al., 2010a*). In agreement with previous reports (*Gonzales et al., 2010b*; *Gonzales et al., 2010a*), we found that TICC activity in VSMCs from control mice was increased following application of negative pressure (–20 mmHg) through the patch pipette to stretch the PM, an effect that was attenuated by the selective TRPM4 blocker, 9-phenanthrol (*Figure 7F*). TICC activity and amplitude in VSMCs isolated from *Stim1*-smKO mice were significantly reduced compared with controls (*Figure 7F–H*). To determine if these differences were attributable to changes in TRPM4 channel function or availability, we activated TRPM4 currents in VSMCs from *Stim1*-smKO and control mice using an internal solution containing 200 µM free $Ca^{2+}$ and compared whole-cell TRPM4 currents in both groups by patch-clamping VSMCs in the conventional whole-cell configuration (*Amarouch et al., 2013*). The TRPM4-sensitive component of the current was isolated by applying 9-phenanthrol. We found that whole-cell TRPM4 current amplitudes did not differ between VSMCs from control and *Stim1*-smKO mice (*Figure 7I and J*), suggesting that the number of TRPM4 channels available for activation at the PM was not altered by *Stim1* knockout. *Stim1* knockout did not alter mRNA levels of TRPM4 subunits or any of the $IP_3R$ subtypes (1, 2, or 3) in cerebral arteries (*Figure 7—figure supplement 2A*). In addition, blockade of Orai1 had no effect on TICC activity (*Figure 7—figure supplement 2B and C*). These findings suggest that diminished TICC activity following knockout of *Stim1* is not due to diminished expression of TRPM4 or $IP_3Rs$, and that generation of TICCs is independent of Orai1 channel activity.

## The contractility of resistance arteries from *Stim1*-smKO mice is blunted

Knockout of *Stim1* in VSMCs decreased the activity of BK and TRPM4 channels under physiological recording conditions. These channels have opposing effects on VSMC membrane potential, contractility, and arterial diameter, with BK channels causing dilation (*Nelson et al., 1995*) and TRPM4 channels causing constriction (*Earley et al., 2004*). Thus, the overall functional impact of deficient channel activity is not immediately apparent. Therefore, to investigate the net consequences of *Stim1* knockout on arterial contractile function, we employed a series of ex vivo pressure myography experiments. Constrictions of intact cerebral pial arteries in response to a depolarizing concentration (60 mM) of extracellular KCl did not differ between groups (*Figure 8A*), suggesting that knocking out *Stim1* in cerebral artery SMCs did not grossly alter voltage-dependent $Ca^{2+}$ influx or underlying contractile

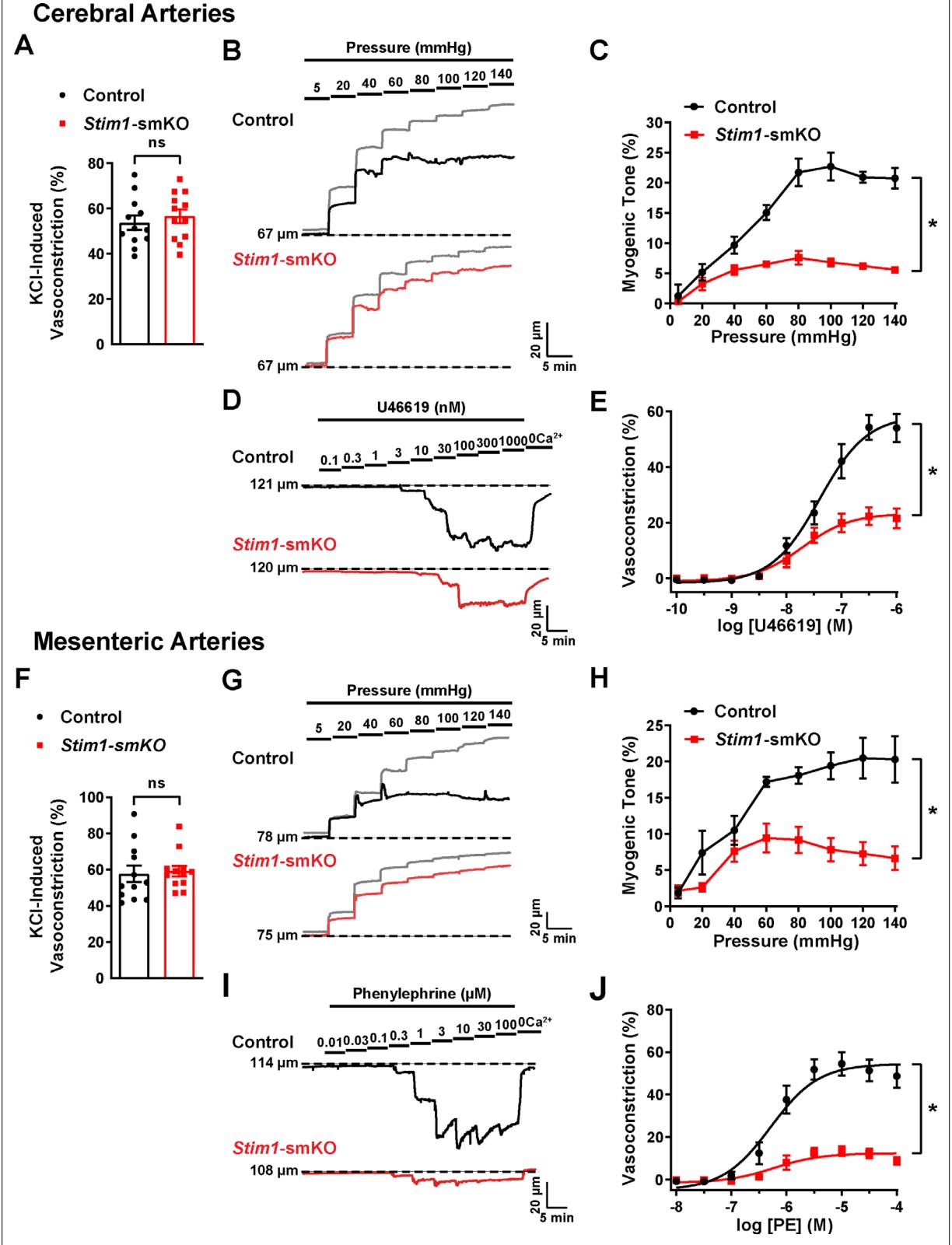

**Figure 8.** Resistance arteries from *Stim1*-smKO mice are dysfunctional. (**A**) Summary data showing vasoconstriction of cerebral pial arteries isolated from control and *Stim1*-smKO mice in response to 60 mM KCl (n = 12 vessels from six mice for both groups, unpaired *t*-test). ns, not significant. (**B**) Representative traces showing changes in luminal diameter over a range of intraluminal pressures (5–140 mmHg) in cerebral pial arteries isolated from control (black trace) and *Stim1*-smKO (red) mice. Gray traces represent passive responses (Ca²⁺-free solution) to changes in intraluminal pressure

*Figure 8 continued on next page*

*Figure 8 continued*

for each artery. (**C**) Summary data showing myogenic reactivity as a function of intraluminal pressure (n = 6 vessels from three mice for each group; *p<0.05, two-way ANOVA). (**D**) Representative traces showing changes in luminal diameter in response to a range of concentrations (0.1–1,000 nM) of the vasoconstrictor agonist U46619 in cerebral arteries isolated from control (black trace) and *Stim1*-smKO (red trace) mice. (**E**) Summary data showing vasoconstriction as a function of U46619 concentration (n = 6 vessels from three mice for each group; *p<0.05, two-way ANOVA). (**F**) Summary data showing vasoconstriction of third-order mesenteric arteries isolated from control and *Stim1*-smKO mice in response to 60 mM KCl (n = 12 vessels from six mice for both groups, unpaired *t*-test). ns, not significant. (**G**) Representative traces showing changes in luminal diameter over a range of intraluminal pressures (5–140 mmHg) in third-order mesenteric arteries isolated from control (black trace) and *Stim1*-smKO (red) mice. Gray traces represent passive responses to changes in intraluminal pressure for each artery. (**H**) Summary data for myogenic reactivity as a function of intraluminal pressure (n = 6 vessels from three mice for each group, *p<0.05, two-way ANOVA). (**I**) Representative traces showing changes in luminal diameter in response to a range of concentrations (0.01–100 μM) of the vasoconstrictor agonist phenylephrine (PE) in third-order mesenteric arteries isolated from control (black trace) and *Stim1*-smKO (red trace) mice. (**J**) Summary data for vasoconstriction as a function of PE concentration, presented as means ± SEM (n = 6 vessels from three mice for each group; *p<0.05, two-way ANOVA).

The online version of this article includes the following source data and figure supplement(s) for figure 8:

**Source data 1.** Individual data points and analysis summaries for datasets shown in *Figure 8*.

**Figure supplement 1.** Effects of pharmacological inhibition of BK channel, TRPM4 channel and Orai1 channel on myogenic tone in isolated cerebral arteries.

**Figure supplement 1—source data 1.** Individual data points and analysis summaries for datasets shown in *Figure 8—figure supplement 1*.

processes. Contractile responses to increases in intraluminal pressure (myogenic vasoconstriction) were evaluated by measuring steady-state luminal diameter at intraluminal pressures over a range of 5–140 mmHg in the presence (active response) and absence (passive response) of extracellular $Ca^{2+}$. Myogenic tone, calculated by normalizing active constriction to passive dilation, was significantly lower in cerebral arteries from *Stim1*-smKO mice compared with those from controls (*Figure 8B and C*). Contractile responses to the synthetic thromboxane $A_2$ receptor agonist U46619 were also significantly blunted in cerebral arteries from *Stim1*-smKO mice compared with those from vehicle-treated controls (*Figure 8D and E*). These data demonstrate that the ability of cerebral arteries from *Stim1*-smKO mice to contract in response to physiological stimuli is impaired. Additional investigations using third-order mesenteric arteries yielded similar findings (*Figure 8F–J*), indicating widespread vascular dysfunction in *Stim1*-smKO mice.

Further experiments investigated the effects of the BK channel inhibitor paxilline and the TRPM4 channel inhibitor 9-phenanthrol on vasoconstriction of cerebral arteries isolated from control and *Stim1*-smKO mice. We found that paxilline increased myogenic tone in cerebral arteries isolated from control mice, whereas this treatment had little effect on cerebral arteries from *Stim1*-smKO mice (*Figure 8—figure supplement 1A and B*). These findings are consistent with the patch-clamp electrophysiology data indicating low levels of BK channel activity in VSMCs from *Stim1*-smKO mice. Treatment with 9-phenanthrol abolished the myogenic tone of cerebral arteries from control mice but had little effect on cerebral arteries from *Stim1*-smKO mice (*Figure 8—figure supplement 1C and D*), in agreement with the patch-clamp electrophysiology studies that found low levels of TICC activity in VSMCs from *Stim1*-smKO mice.

We also found that Synta66 had no effect on KCl-induced vasoconstriction or myogenic tone of cerebral arteries from control mice, indicating that these responses are independent of Orai1 channel activity (*Figure 8—figure supplement 1E–G*).

## *Stim1*-smKO mice are hypotensive

Age-matched $Myh11^{CreERT2}$: $Stim1^{fl/fl}$ mice were surgically implanted with radio telemetry transmitters as previously described (*Li et al., 2014*). After a recovery period (14 days), systolic and diastolic blood pressure (BP), heart rate (HR), and locomotor activity levels were recorded for 48 hr before tamoxifen injection (control). Systolic and diastolic BP, HR, and activity levels were again recorded for 48 hr, beginning 1 week after completing the tamoxifen injection protocol (*Stim1*-smKO). Normal diurnal variations were observed for all parameters (*Figure 9*). The mean systolic BP of *Stim1*-smKO mice was lower than that of control mice during both day and night cycles (*Figure 9A*), whereas diastolic BP did not differ between groups (*Figure 9B*). Mean arterial pressure (MAP) (*Figure 9C*) was lower in *Stim1*-smKO mice compared with controls at night and tends to be lower during the day (p=0.056). The pulse pressure of *Stim1*-smKO mice was lower than that of control mice during both day and night

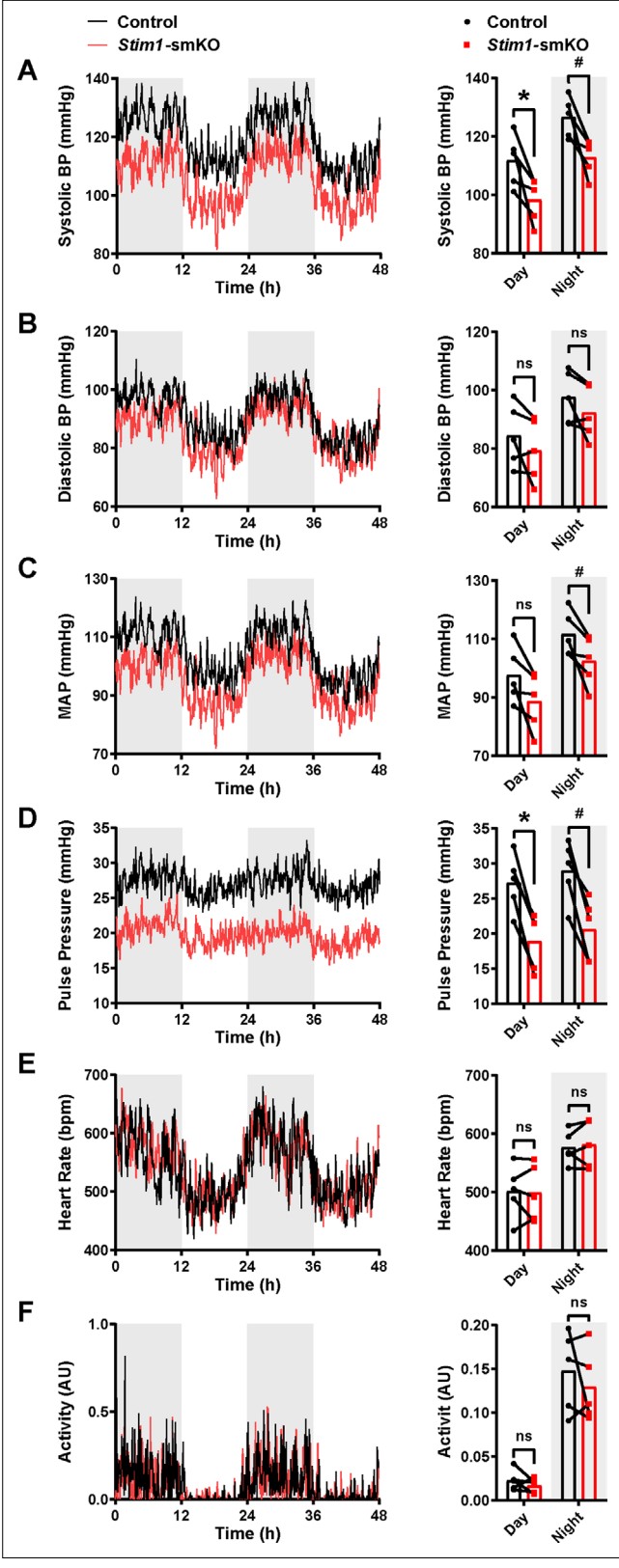

**Figure 9.** *Stim1*-smKO mice are hypotensive. (**A**) Systolic blood pressure (BP) (mmHg) over 48 hr in conscious, radio telemeter-implanted *Myh11CreERT2: Stim1fl/fl* mice before (control) and after (*Stim1*-smKO) tamoxifen injection. Shaded regions depict night cycles (n = 5 for both groups; *p<0.05 vs. control day, #p<0.05 vs. control night, paired *t*-test). (**B**) Diastolic BP measurements for control and *Stim1*-smKO mice (n = 5 for both groups, paired

*Figure 9 continued on next page*

*Figure 9 continued*

*t*-test). ns, not significant. (**C**) Mean arterial pressure (MAP) for control and *Stim1*-smKO mice (n = 5 for both groups, #p<0.05 vs. control night, paired *t*-test). ns, not significant. (**D**) Pulse pressure for control and *Stim1*-smKO mice (n = 5 for both groups; *p<0.05 vs. control day, #p<0.05 vs. control night, paired *t*-test). (**E**) Heart rate (HR) for control and *Stim1*-smKO mice (n = 5 for both groups, paired *t*-test). ns, not significant. (**F**) Locomotor activity (arbitrary units [AU]) for control and *Stim1*-smKO mice (n = 5 for both groups, paired *t*-test). ns, not significant. 48 hr recordings are shown as means; bar graphs are shown as means ± SEM.

The online version of this article includes the following source data and figure supplement(s) for figure 9:

**Source data 1.** Individual data points and analysis summaries for datasets shown in *Figure 9*.

**Figure supplement 1.** Blood pressure, heart rate and activity remain unaltered in *Myh11*<sup>CreERT2</sup>: *Stim1*<sup>fl/fl</sup> mice after vehicle injection.

**Figure supplement 1—source data 1.** Individual data points and analysis summaries for datasets shown in *Figure 9—figure supplement 1*.

---

cycles (*Figure 9D*). HR (*Figure 9E*), and locomotor activity (*Figure 9F*) did not differ between groups. Vehicle injection did not affect BP, HR, or locomotor activity (*Figure 9—figure supplement 1*). These data indicate that acute knockout of *Stim1* in VSMCs lowers BP, probably due to diminished arterial contractility and decreased total peripheral resistance.

## Discussion

Junctional membrane complexes formed by close interactions of the ER/SR with the PM are critical signaling hubs that regulate homeostatic and adaptive processes in nearly every cell type. The canonical function of STIM1 is to enable SOCE via Orai channels, but mounting evidence suggests that the protein has additional, SOCE-independent functions. Here, we show that STIM1 is crucial for fostering SR-PM junctions and functional coupling between SR and PM ion channels that control VSMC contractility. In support of this concept, we found that the number and sizes of SR/PM coupling sites were significantly reduced in VSMCs from *Stim1*-smKO mice. *Stim1* knockout also altered the nanoscale architecture of ion channels in $Ca^{2+}$-signaling complexes, transformed the properties of $Ca^{2+}$ sparks, and diminished BK and TRPM4 channel activity under physiological recording conditions. BK and TRPM4 channel activity and vasoconstrictor responsiveness were not altered by selective inhibition of Orai1. Resistance arteries isolated from *Stim1*-smKO mice exhibited blunted responses to vasoconstrictor stimuli, and animals became hypotensive following acute knockout of *Stim1* in smooth muscle. These findings collectively demonstrate that in contractile VSMCs STIM1 expression is necessary for the functional coupling of $Ca^{2+}$ release sites on the SR and $Ca^{2+}$-activated ion channels on the PM in a manner independent of Orai1, $Ca^{2+}$ store depletion, and SOCE. Loss of functional coupling in VSMCs following *Stim1* knockout has profound consequences, disrupting arterial function and BP regulation.

The SR-PM signaling domains of VSMCs are less orderly compared with those in cardiac and skeletal muscle cells and remain incompletely characterized. SR-PM junctions within the transverse (T) tubules of cardiomyocytes and skeletal muscle cells have regular, repeating structures that are formed, in part, by cytoskeletal elements and proteins of the junctophilin (*Beavers et al., 2014*; *Landstrom et al., 2014*; *Takeshima et al., 2000*) and triadin (*Knudson et al., 1993*; *Marty, 2015*) families. In VSMCs, which lack T-tubules, SR-PM interactions occur at peripheral coupling sites that form throughout the periphery with no apparent distribution pattern. Our research team has previously identified vital roles for microtubule networks (*Pritchard et al., 2017*) and junctophilin 2 (JPH2) (*Pritchard et al., 2019*) in the formation of peripheral coupling sites in VSMCs. Here, we found that knockout of *Stim1* in VSMCs with intact SR $Ca^{2+}$ stores reduced the number and sizes of SR-PM colocalization sites. Why is STIM1 active under these conditions? A simple explanation is that resting SR $[Ca^{2+}]$ in fully differentiated, contractile VSMCs is sufficiently low to trigger constitutive activation of STIM1. This concept is supported by a report by *Luik et al., 2008*, who showed that the half-maximal concentration ($K_{1/2}$) of ER $Ca^{2+}$ for the activation of $I_{CRAC}$ in Jurkat T cells is 169 μM and the $K_{1/2}$ for redistribution of STIM1 to the PM is 187 μM. These data are in close agreement with another study, which reported that the $K_{1/2}$ of ER $Ca^{2+}$ for redistribution of STIM1 in HeLa cells was 210 μM and that for maximum redistribution was 150 μM (*Brandman et al., 2007*). Few studies have reported SR $[Ca^{2+}]$ measurements in native, contractile VSMCs. Using the low-affinity ratiometric $Ca^{2+}$ indicator, mag-fura-2, one well-controlled

study estimated that resting SR [$Ca^{2+}$] in contractile SMCs was ~110 μM (*ZhuGe et al., 1999*). Under these conditions, STIM1 is expected to be in a fully active configuration that is also supported by our data where thapsigargin failed to increase the number or size of STIM1 puncta in contractile VSMCs. It is also possible that regional SR [$Ca^{2+}$] levels near active $Ca^{2+}$-release sites (RyRs and $IP_3R$) are lower than global SR [$Ca^{2+}$], which could further stimulate STIM1 activity at these sites and reinforce junctional coupling. Thus, we put forward the concept that STIM1 is in an active state in quiescent contractile smooth muscle and is necessary for forming $Ca^{2+}$-signaling complexes that are vital for contractile function. Our data further imply that, as VSMCs transition to a proliferative phenotype during the development of disease states associated with vascular remodeling, SR $Ca^{2+}$ levels increase, leading to STIM1 inactivation, loss of stable peripheral coupling, and acquisition of SOCE activity (*Zhang et al., 2011*).

It would be useful to define the molecular composition of microdomains formed by the interactions of the SR and PM in VSMCs. However, we cannot image SR and PM dyes in native cells using GSDIM because the high laser levels and long exposure times required for this technique bleach the dyes. The SIM mode of our LLS instrument is ideal for imaging the dyes (due to low bleaching) but lacks the resolution of the GSDIM system needed for the nanoscale detection of protein clusters. Consequently, we cannot simultaneously image the sites of membrane interaction and protein clusters. An examination of our superresolution maps suggests that all protein clusters are uniformly distributed – there are no apparent sights of enrichment. Therefore, it seems likely that the ion channel content of the SR:PM interacting domains does not significantly differ from regions of the PM that do not interact with the SR.

Ion channel proteins in the membranes of excitable cells form discreet clusters whose sizes are exponentially distributed, a phenomenon that has been suggested to occur through stochastic self-assembly (*Sato et al., 2019*). Here, we found that acute knockout of STIM1 in VSMCs reduced the mean sizes of BK, TRPM4, and $IP_3R$ protein clusters and slightly increased the mean size of RyR2 protein clusters. According to the stochastic model proposed by *Sato et al., 2019*, the steady-state size of membrane protein clusters is limited by the probability of removal from the PM through recycling or degradation processes, with larger clusters having a higher likelihood of removal. Thus, the smaller size of BK, TRPM4, and $IP_3R$ clusters following STIM1 knockout is likely a consequence of an increase in the rate of channel removal from the membrane. STIM1 knockout also caused a reduction in the SR volume as it retracts from the PM, suggesting that STIM1 is necessary for maintaining contact between the peripheral SR and PM. Accordingly, we propose that by maintaining a connection between the peripheral SR and PM, STIM1 increases the dwell time of BK, TRPM4, and $IP_3Rs$ proteins in the membrane, allowing larger clusters to form. This could be the result of direct protein-protein interactions. For example, previous studies have provided evidence of direct interactions between STIM1 and $IP_3Rs$ (*Béliveau et al., 2014*; *Sampieri et al., 2018*), and our data show that STIM1 interacts with BK and TRPM4 at the nanoscale, potentially influencing cluster formation. However, our data show that the majority of BK and TRPM4 protein clusters do not colocalize with STIM1. It is more likely that the intact peripheral SR partially protects membrane proteins from endocytic and/or recycling cascades, allowing larger clusters to form before they are removed. Loss of the peripheral SR following STOM1 knockdown removes this defense, decreasing the dwell time of proteins in the PM, resulting in smaller clusters.

Knockout of *Stim1* in VSMCs significantly impacted $Ca^{2+}$ signaling, ion channel activity, vascular contractility, and the regulation of BP. We purport that these outcomes result from nanoscale disruptions in cellular architecture. $Ca^{2+}$ sparks occur within microdomains formed by SR/PM junctional sites. An increase in the distance between the two membranes will enlarge the area of the microdomains. This likely explains the observed increase in the spatial spread of $Ca^{2+}$ sparks when STIM1 is knocked out. An enlargement of the microdomains also increases the distance between the source of the $Ca^{2+}$ spark and the SERCA and PMCA pumps and Na/$Ca^{2+}$ exchangers, which remove $Ca^{2+}$ from the cytosol (*Bautista and Lewis, 2004*; *Blaustein and Lederer, 1999*; *Shmigol et al., 1999*), potentially contributing to prolonged decay and increased amplitude seen in cells from *Stim1*-smKO mice. Thus, the compromised structural integrity of subcellular $Ca^{2+}$-signaling microdomains formed by interactions of the PM and SR likely accounts for the altered properties of $Ca^{2+}$ sparks associated with STIM1 knockout. Decreased nanoscale colocalization of BK with RyR2 and TRPM4 with $IP_3Rs$ manifested as diminished $Ca^{2+}$-dependent activity of BK and TRPM4 channels (STOCs and TICCs),

reflecting a loss in the functional coupling of $Ca^{2+}$-release sites on the SR and ion channels on the PM. The smaller sizes of BK and TRPM4 protein clusters on the PM following *Stim1* knockdown may also reduce BK and TRPM4 channel currents. At the intact blood vessel level, the diminished TRPM4 and BK channel activity resulted in impaired contractility in response to physiological stimuli. This finding is interesting because our prior studies investigating the role of microtubular structures (*Pritchard et al., 2017*) and JPH2 *Pritchard et al., 2019* in maintaining peripheral coupling in VSMCs showed that disruption of PM-SR interactions caused cerebral arteries to become hypercontractile. In these studies, arterial hypercontractility resulted from interruption of the BK-RyR2 signaling pathway, which hyperpolarizes the VSMC membrane and balances the depolarizing and contractile influences of the TRPM4-$IP_3$R cascade. *Stim1* knockout, in contrast, affected both pathways, indicating that STIM1 influences peripheral coupling in a manner that differs from that of the microtubule network and JPH2 and further suggesting heterogeneity in the formation of junctional membrane complexes in VSMCs. Diminished arterial contractility following *Stim1* knockout resulted in a drop in arterial BP, probably due to decreased total peripheral resistance. This finding differs from previous reports by other groups showing that, although myogenic tone and phenylephrine-induced vasoconstriction were blunted in mesenteric arteries from a constitutive SMC-specific STIM1-knockout model, resting BP was not affected in this model (*Kassan et al., 2015*; *Kassan et al., 2016*; *Pichavaram et al., 2018*). This difference is likely due to elevated levels of circulating catecholamines, which increase HR and cardiac output and thereby compensate for diminished vascular resistance (*Pichavaram et al., 2018*).

In summary, our data demonstrate a vital role for STIM1 in the maintenance of critical $Ca^{2+}$-signaling microdomains in contractile VSMCs that is independent of SR $Ca^{2+}$ store depletion. Disruptions in cellular architecture at the nanoscale level associated with the loss of STIM1 resulted in arterial dysfunction and impaired BP regulation, highlighting the essential nature of $Ca^{2+}$-signaling complexes formed by SR-PM interactions in cardiovascular control.

## Materials and methods

### Animals

All animal studies were performed in accordance with guidelines of the Institutional Animal Care and Use Committee (IACUC) of the University of Nevada, Reno. Mice were housed in cages on a 12 hr/12 hr day-night cycle with ad libitum access to food (standard chow) and water. All transgenic mouse strains were obtained from The Jackson Laboratory (Bar Harbor, ME). Mice with *loxP* sites flanking exon 2 of the *Stim1* gene (*Stim1*$^{fl/fl}$ mice) were crossed with myosin heavy chain 11 *Myh11*$^{CreERT2}$ mice (*Chappell et al., 2016*; *Wirth et al., 2008*), generating *Myh11*$^{CreERT2}$: *Stim1*$^{fl/wt}$ mice. Heterozygous *Myh11*$^{CreERT2}$: *Stim1*$^{fl/wt}$ mice were then intercrossed, yielding *Myh11*$^{CreERT2}$: *Stim1*$^{fl/fl}$ mice.

### Induction of STIM1 knockout

Male *Myh11*$^{CreERT2}$: *Stim1*$^{fl/fl}$ mice were intraperitoneally injected at 4–6 weeks of age with 100 µL of a 10 mg/mL tamoxifen solution once daily for 5 days to produce *Stim1*-smKO mice. Mice were used for experiments 1 week after the final injection. Littermate *Myh11*$^{CreERT2}$: *Stim1*$^{fl/fl}$ mice injected with the vehicle for tamoxifen (sunflower oil) were used as controls for all experiments.

### Wes capillary electrophoresis

Tissues isolated from mice were homogenized in ice-cold RIPA buffer (25 mM Tris pH 7.6, 150 mM NaCl, 1% Igepal CA-630, 1% sodium deoxycholate, 0.1% SDS) with protease inhibitor cocktail (Cell Biolabs, Inc, San Diego, CA) using a mechanical homogenizer followed by sonication. The resulting homogenate was centrifuged at 14,000 rpm for 20 min at 4°C, and the supernatant containing proteins was collected. Protein concentration was quantified with a BCA protein assay kit (Thermo Scientific, Waltham, MA) by absorbance spectroscopy using a 96-well plate reader. Proteins were then resolved by capillary electrophoresis using the Wes system (ProteinSimple, San Jose, CA) and probed with an anti-STIM1 primary antibody (S6072; Sigma-Aldrich, St. Louis, MO). Total protein expression was quantified using the *Total Protein Detection Module for Wes* from ProteinSimple, which utilizes biotin labeling of all proteins that are then detected using Streptavidin-HRP chemiluminescence. Bands were analyzed using Compass for SW (ProteinSimple). STIM1 band intensities were normalized to the total protein band intensities of the respective samples.

## SMC isolation

Mice were euthanized by decapitation and exsanguination under isoflurane anesthesia. Cerebral pial arteries were isolated carefully in ice-cold $Mg^{2+}$-containing physiological salt solution ($Mg^{2+}$-PSS; 5 mM KCl, 140 mM NaCl, 2 mM $MgCl_2$, 10 mM HEPES, and 10 mM glucose; pH 7.4, adjusted with NaOH) and then incubated in an enzyme cocktail containing 1 mg/mL papain (Worthington Biochemical Corp., Lakewood, NJ), 1 mg/mL dithiothreitol (DTT; Sigma-Aldrich), and 10 mg/mL bovine serum albumin (BSA; Sigma-Aldrich) for 12 min at 37°C. The arteries were then washed three times with $Mg^{2+}$-PSS and incubated in 1 mg/mL collagenase type II (Worthington) in $Mg^{2+}$-PSS for 14 min. The arteries were washed three times with $Mg^{2+}$-PSS and then dissociated into single cells by triturating with a fire-polished glass Pasteur pipette.

## SOCE

Contractile and proliferative VSMCs were allowed to attach to glass coverslips overnight, and coverslips were then mounted to a Teflon chamber and incubated with 4 µM Fura-2 AM in complete media at 37°C for 45 min. Loading solution included 0.1% of Pluronic F-127 to facilitate Fura-2 AM loading. Cells were then washed with a HEPES-buffered salt solution (HBSS) containing 140 mM NaCl, 4.7 mM KCl, 1.13 mM $MgCl_2$, 10 mM HEPES, 2.0 mM $CaCl_2$, and 10 mM glucose (pH 7.4 adjusted by NaOH). Cells were then incubated for 10 min in HBSS at room temperature before recordings. Coverslips were then mounted to Nikon TS100 inverted microscope equipped with a 20× Fluor objective and 0.75 numerical aperture. Fura-2 AM was alternately excited at 340 and 380 nm, and fluorescent emission was captured at 510 nm. Fluorescence from multiple cells (*Mercer et al., 2006*; *Kwon et al., 2017*; *Prakriya and Lewis, 2015*; *Emrich et al., 2022*; *Abdullaev et al., 2008*; *Berry et al., 2011*; *Chin-Smith et al., 2014*; *Correll et al., 2015*; *Jones et al., 2008*; *Klejman et al., 2009*; *Koh et al., 2009*; *López et al., 2008*; *Lu et al., 2010*; *Lu et al., 2008*; *Lyfenko and Dirksen, 2008*; *Numaga-Tomita and Putney, 2013*; *Nurbaeva et al., 2015*; *Onodera et al., 2013*; *Peel et al., 2006*; *Takahashi et al., 2007*; *Wissenbach et al., 2007*; *Zhang et al., 2007*; *Zhou et al., 2015*; *Perni et al., 2015*; *Soboloff et al., 2006*; *Spassova et al., 2006*) were recorded and analyzed with a digital fluorescence imaging system (InCytim2, Intracellular Imaging, Cincinnati, OH). The fluorescence ratio at 340 and to 380 was obtained for each pixel. Thapsigargin at a final concentration of 2 µM was suspended in HBSS. Mean data are reported as the peak $F_{340}/F_{380}$ ratio.

## Visualization of PM-SR colocalization sites using SIM

Cerebral pial artery SMCs were allowed to adhere onto poly-L-lysine-coated round coverslips (5 mm diameter) during a 30 min incubation at 37°C with the SR stain, ER-Tracker Green (Thermo Fisher Scientific; diluted 1:1000 in $Mg^{2+}$-PSS). After incubation, ER-Tracker Green was removed, and the PM stain Cell-Mask Deep Red (Thermo Fisher Scientific; diluted 1:1000 in $Mg^{2+}$-PSS) was added, and cells were incubated for 5 min at 37°C. Cell-Mask Deep Red was then removed, and cells were washed with $Mg^{2+}$-PSS and imaged using a lattice light-sheet microscope (LLSM; Intelligent Imaging Innovations, Inc, Denver, CO) (*Chen et al., 2014*). Coverslips with stained cells were mounted onto a sample holder and placed in the LLSM bath, immersed in $Mg^{2+}$-PSS. Imaging was performed in SR-SIM mode, set to 100 ms exposures. For each cell, 200 Z-steps were collected at a step size of 0.25 µm. Imaging was limited to no more than 30 min for each coverslip to prevent artifacts caused by internalization of the PM dye. Surface reconstruction and colocalization analyses of PM and SR were performed using Imaris v9.8 (Bitplane, Zurich, Switzerland) image analysis software. The Surface-Surface coloc plug-in was used to visualize areas of the PM and SR that colocalized to form coupling sites. PM-SR colocalization percentage was calculated by dividing the total PM-SR colocalization site volume by PM volume and multiplying by 100. Using fluorescent beads, we determined that for the SIM modality of the LLS the resolution for 642 nm wavelength (used for PM labeling) is 250–335 nm and the resolution for 488 nm wavelength (used for SR labeling) is 225–295 nm.

## GSDIM superresolution microscopy

GSDIM was performed as described previously (*Pritchard et al., 2019*; *Thakore, 2020*; *Pritchard et al., 2017*; *Griffin et al., 2020*; *Pritchard et al., 2018*). For epifluorescence imaging, freshly isolated cerebral pial artery SMCs were allowed to adhere onto poly-L-lysine-coated glass coverslips for 30 min. For TIRF imaging, the coverslips were first cleaned by sonicating in 5 N NaOH for 45 min

and then sonicated for another 45 min in deionized water before adding freshly isolated cerebral pial artery SMCs. The cells were then fixed for 20 min with 2% paraformaldehyde, quenched with 0.4 mg/mL NaBH$_4$, and permeabilized with 0.1% Triton X-100. Cells were then blocked with 50% SEABLOCK blocking buffer (Thermo Fisher Scientific) for 2 hr and incubated overnight at 4°C with primary antibody (anti-STIM1- [4916], Cell Signaling Technologies, Danvers, MA; anti-STIM1 [610954], BD Biosciences, Franklin Lakes, NJ; anti-BKα1- [APC-021], Alomone Labs, Jerusalem, Israel; anti-RyR2- [MA3-916], Thermo Fisher Scientific; anti-TRPM4- (ABIN572220); https://antibodies-online.com, Limerick, PA; anti-IP$_3$R- (ab5804), Abcam, Cambridge, UK) diluted in PBS containing 20% SEABLOCK, 1% BSA, and 0.05% Triton X-100. Cells were washed three times with 1× PBS after each step. After overnight incubation, unbound primary antibody was removed by washing four times with 20% SEABLOCK, after which cells were incubated with secondary antibodies (Alexa Fluor 647- or Alexa Fluor 568-conjugated goat anti-rabbit, goat anti-mouse, donkey anti-goat, or donkey anti-rabbit as appropriate) at room temperature for 2 hr in the dark. After washing with 1× PBS, coverslip-plated cells were mounted onto glass depression slides in a thiol-based photo-switching imaging buffer consisting of 50 mM Tris/10 mM NaCl (pH 8), 10% glucose, 10 mM mercaptoethylamine, 0.48 mg/mL glucose oxidase, and 58 µg/mL catalase. Coverslips were sealed to depression slides with Twinsil dental glue (Picodent, Wipperfurth, Germany) to exclude oxygen and prevent rapid oxidation of the imaging buffer. Superresolution images were acquired in epifluorescence or TIRF mode using a GSDIM imaging system (Leica, Wetzlar, Germany) equipped with an oil-immersion 160× HCX Plan-Apochromat (NA 1.47) objective, an electron-multiplying charge-coupled device camera (EMCCD; iXon3 897; Andor Technology, Belfast, UK), and 500 mW, 532- and 642 nm laser lines. Localization maps were constructed from images acquired at 100 Hz for 25,000 frames using Leica LAX software. Post-acquisition image analyses of cluster size distribution were performed using binary masks of images in NIH ImageJ software. OBA was used to establish colocalization of proteins of interest in superresolution localization maps.

## Object-based colocalization analysis

OBA was used to establish the colocalization of BK channels with RyR2 and TRPM4 channels with IP$_3$R in superresolution localization maps. We used NIH ImageJ software with the JACoP colocalization analysis plug-in (*Bolte and Cordelières, 2006*; *Lachmanovich et al., 2003*). The JACoP plug-in was used to split the two channels representing fluorophores detected by Alexa Fluor 568 or Alexa Fluor 647. Contiguous objects in both channels were identified by systematically inspecting the neighboring eight pixels (in 2D) of a reference pixel. All adjacent pixels with intensities above a user-defined threshold limit were considered part of the same structure as the reference pixel and were segmented as individual objects. The superresolution localization maps were previously thresholded by the detection algorithm incorporated into the LAX software used for image acquisition. Therefore, the threshold level in JACoP was set to 1 (nearly the minimum) for all images. After thresholding, centroids (defined as the single-pixel geometric centers of the specified objects) were determined for each object. Clusters in the other wavelength within 20 nm (resolution limit of our GSDIM system) of the centroid were considered 'colocalized.' The percentage of colocalizing clusters was calculated as the number of colocalizing clusters as a percentage divided by the total number of clusters detected. This method of colocalization analysis can overcome artifacts caused due to uneven fluorescence intensities and is appropriate to use when the analyzed objects in question are small and punctate like the protein clusters in our images (*Lachmanovich et al., 2003*). For comparison, new maps were generated from each original superresolution map using JACoP that replicated the density and cluster size distribution, but the location of each protein cluster was assigned to a random site. We then performed OBA for the simulated random distribution and compared the colocalization frequency of the original maps with the colocalization frequency of their randomized counterparts.

## Patch-clamp electrophysiology

Freshly isolated cerebral artery SMCs were transferred to the recording chamber and allowed to adhere to glass coverslips at room temperature for 20 min. Recording electrodes (3–4 MΩ) were pulled on a model P-87 micropipette puller (Sutter Instruments, Novado, CA) and polished using a MF-830 MicroForge (Narishige Scientific Instruments Laboratories, Long Island, NY). STOCs and TICCs were recorded in Ca$^{2+}$-containing PSS (134 mM NaCl, 6 mM KCl, 1 mM MgCl$_2$, 2 mM CaCl$_2$,

10 mM HEPES, and 10 mM glucose; pH 7.4, adjusted with NaOH). The patch pipette solution contained 110 mM K-aspartate, 1 mM $MgCl_2$, 30 mM KCl, 10 mM NaCl, 10 mM HEPES, and 5 μM EGTA (pH 7.2, adjusted with NaOH). Amphotericin B (200 μM), prepared on the day of the experiment, was included in the pipette solution to perforate the membrane. Currents were recorded using an Axopatch 200B amplifier equipped with an Axon CV 203BU headstage (Molecular Devices) for all experiments. Currents were filtered at 1 kHz, digitized at 40 kHz, and stored for subsequent analysis. Clampex and Clampfit (version 10.2; Molecular Devices) were used for data acquisition and analysis, respectively. For STOCs, cells were clamped at a membrane potential manually spanning a range from –60 mV to 0 mV. STOCs were defined as events >10 pA, and their frequency was calculated by dividing the number of events by the time between the first and last event. The potential contribution of Orai1 channels to STOCs was assessed by applying Synta66 (10 μM) to the bath solution, while STOCs were recorded at a physiological membrane potential (–40 mV). Whole-cell $K^+$ currents were recorded using a step protocol (–100 to +100 mV in 20 mV steps for 500 ms) from a holding potential of –80 mV. Whole-cell BK currents were calculated by current subtraction following administration of the selective BK channel blocker paxilline (1 μM). Current-voltage (I–V) plots were generated using currents averaged over the last 50 ms of each voltage step. The bathing solution contained 134 mM NaCl, 6 mM KCl, 10 mM HEPES, 10 mM glucose, 2 mM $CaCl_2$, and 1 mM $MgCl_2$; pH 7.4 (NaOH). The pipette solution contained 140 mM KCl, 1.9 mM $MgCl_2$, 75 μM $Ca^{2+}$, 10 mM HEPES, 0.1 mM EGTA, and 2 mM $Na_2ATP$; pH 7.2 (KOH).

TICCs, induced by membrane stretch delivered by applying negative pressure (20 mmHg) through the recording electrode using a Clampex controlled pressure clamp HSPC-1 device (ALA Scientific Instruments Inc, Farmingdale, NY, USA), were recorded from cells clamped at a membrane potential of –70 mV. TICC activity was calculated as the sum of the open channel probability (NPo) of multiple 1.75 pA open states (*Gonzales et al., 2010b*). The contribution of Orai1 channels to TICCs activity was assessed by applying Synta66 (10 μM) in the bath solution after activating TICCs through membrane stretch. Conventional whole-cell TRPM4 currents were recorded using ramp protocol consisting of a 400 ms increasing ramp from –100 to +100 mV ending with 300 ms step at +100 mV from a holding potential of –60 mV. A new ramp was applied every 2 s. TRPM4 whole-cell currents were recorded in a bath solution consisting of (in mM) 156 NaCl, 1.5 $CaCl_2$, 10 glucose, 10 HEPES, and 10 TEA-Cl; pH 7.4 (NaOH). The patch pipette solution contained (in mM) 156 CsCl, 8 NaCl, 1 $MgCl_2$ 10 mM HEPES; pH 7.4 (NaOH) and 200 μM free $[Ca^{2+}]$, adjusted with an appropriate amount of $CaCl_2$ and EGTA as calculated using Max-Chelator software.

## Quantitative droplet digital PCR

Total RNA was extracted from arteries by homogenization in TRIzol reagent (Invitrogen, Carlsbad, CA), followed by purification using a Direct-zol RNA microprep kit (Zymo Research, Irvine, CA), DNase I treatment (Thermo Fisher Scientific), and reverse transcription into cDNA using qScript cDNA Supermix (Quanta Biosciences, Gaithersburg, MD). Quantitative droplet digital PCR (ddPCR) was performed using QX200 ddPCR EvaGreen Supermix (Bio-Rad, Hercules, CA), custom-designed primers (*Supplementary file 2*), and cDNA templates. Generated droplet emulsions were amplified using a C1000 Touch Thermal Cycler (Bio-Rad), and the fluorescence intensity of individual droplets was measured using a QX200 Droplet Reader (Bio-Rad) running QuantaSoft (version 1.7.4; Bio-Rad). Analysis was performed using QuantaSoft Analysis Pro (version 1.0.596; Bio-Rad).

## Imaging of $Ca^{2+}$ sparks

A liquid suspension (~0.2 mL) of freshly isolated VSMCs was placed in a recording chamber (RC-26GLP, Warner Instruments, Hamden, CT) and allowed to adhere to glass coverslips for 20 min at room temperature. VSMCs were then loaded with the $Ca^{2+}$-sensitive fluorophore, Fluo-4 AM (1 μM; Molecular Probes), in the dark for 20 min at room temperature in $Mg^{2+}$-PSS. Cells were subsequently washed three times with $Ca^{2+}$-containing PSS and incubated at room temperature for 20 min in the dark to allow sufficient time for Fluo-4 de-esterification. Images were acquired using an iXon 897 EMCCD camera (Andor; 16 × 16 μm pixel size) coupled to a spinning-disk confocal head (CSU-X1; Yokogawa), with a 100× oil-immersion objective (Olympus; NA 1.45) at an acquisition rate of 33 frames per second (fps). Custom software (SparkAn; https://github.com/vesselman/SparkAn)(Dabertrand et al., 2012) provided by Dr. Adrian D. Bonev (University of Vermont) was used to analyze the properties of $Ca^{2+}$

sparks. The threshold for $Ca^{2+}$ spark detection was defined as local increases in fluorescence $\geq 0.2 \Delta F/F_0$.

## Pressure myography

Pressure myography experiments were conducted using current guidelines (*Wenceslau et al., 2021*). Cerebral pial and third-order mesenteric arteries were carefully isolated in ice-cold $Mg^{2+}$-PSS. Each artery was then cannulated and mounted in an arteriography chamber and superfused with oxygenated (21% $O_2$/6% $CO_2$/73% $N_2$) $Ca^{2+}$-PSS (119 mM NaCl, 4.7 mM KCl, 21 mM $NaCO_3$, 1.18 mM $KH_2PO_4$, 1.17 mM $MgSO_4$, 0.026 mM EDTA, 1.8 mM $CaCl_2$, and 4 mM glucose) at 37°C and allowed to stabilize for 15 min. Each artery was then pressurized to 110 mmHg using a pressure servo controller (Living Systems Instruments, St. Albans City, VT). Any kinks or bends were gently straightened out, the pressure was reduced to 5 mmHg, and the artery was allowed to stabilize for 15 min. The viability of each artery was assessed by measuring the response to high extracellular [$K^+$] PSS (made isotonic by adjusting the [NaCl], 60 mM KCl, 63.7 mM NaCl). Arteries that contracted less than 10% were excluded from further investigation.

Myogenic tone was assessed by raising the intraluminal pressure from 5 mmHg to 140 mmHg in 20 mmHg increments, with the artery maintained at each pressure increment for 5 min (active response). The artery was then superfused for 15 min at 5 mmHg intraluminal pressure with $Ca^{2+}$-free PSS supplemented with EGTA (2 mM) and the voltage-dependent $Ca^{2+}$ channel blocker diltiazem (10 µM), followed by application of pressure increments from 5 mmHg to 140 mmHg (passive response). The artery lumen diameter was recorded using edge-detection software (IonOptix, Westwood, MA). Myogenic reactivity at each intraluminal pressure was calculated as [1 − (Active diameter/Passive Diameter)] × 100. The contribution of Orai1 channels to myogenic tone was assessed by treating vessels with Synta66 (10 µM) in the superfusing bath. The effects of BK and TRPM4 channel inhibition on myogenic tone were assessed in vessels pressurized to 60 mmHg. Tone was allowed to develop before administering paxilline (1 µM) in the superfusing bath to inhibit BK channels or 9-phenanthrol (30 µM) to inhibit TRPM4 channels.

In separate arteries, the contractile response to the thromboxane $A_2$ receptor agonist U46619 and $\alpha_1$-adrenergic receptor agonist phenylephrine was assessed in cerebral and mesenteric arteries, respectively. Arteries were pressurized to 20 mmHg to prevent the development of myogenic tone. Cumulative concentration-response curves were produced by adding U46619 (0.01–1000 nM) or phenylephrine (0.001–100 µM) to the superfusing bath solution. Arteries were maintained at each concentration for 5 min or until a steady-state diameter was reached before adding the next concentration. Following the addition of the final concentration, arteries were bathed in $Ca^{2+}$-free PSS to obtain the passive diameter. Contraction was calculated at each concentration as vasoconstriction (%) = [(lumen diameter at constriction − lumen diameter at baseline)/passive lumen diameter] × 100.

## In vivo radiotelemetry

*Stim1*-smKO mice were initially anesthetized using 4–5% isoflurane carried in 100% $O_2$ (flushed at 1 L/min), after which anesthesia was maintained by adjusting isoflurane to 1.5–2%; preoperative analgesia was provided by subcutaneous injection of 50 µg/kg buprenorphine (ZooPharm, Windsor, CO). The neck was shaved and then sterilized with iodine. Under aseptic conditions, an incision (~1 cm) was made to separate the oblique and tracheal muscles and expose the left common carotid artery. The catheter of a radio telemetry transmitter (PA-C10; Data Science International, Harvard Bioscience, Inc, Minneapolis, MN) was surgically implanted in the left common carotid artery and secured using nonabsorbable silk suture threads. The body of the transmitter was embedded in a subcutaneous skin pocket under the right arm. After a 14-day recovery period, baseline BP, HR, and locomotor activity were recorded in conscious mice for 48 hr using Ponemah 6.4 software (Data Science International). Parameters were measured for 20 s every 5 min. Mice were then injected with either vehicle or tamoxifen using the protocol described above; after 7 days following the final injection, baseline BP readings, HR, and locomotor activity were re-recorded in conscious mice for 48 hr.

## Chemicals

Unless specified otherwise, all chemicals used were obtained from Sigma-Aldrich.

## Statistical analysis

All data are expressed as means ± standard error of the mean (SEM) unless specified otherwise. Statistical analyses were performed using paired or unpaired Student's $t$-test, or analysis of variance (ANOVA), as appropriate. A p-value <0.05 was considered to indicate statistically significant differences. GraphPad Prism v9.3 (GraphPad Software, Inc, USA) was used for statistical analyses and graphical presentations.

## Acknowledgements

This study was supported by grants from the National Institutes of Health (NHLBI R35HL155008, R01HL137852, R01HL091905, R01HL139585, R01HL122770, R01HL146054, NINDS RF1NS110044, R61NS115132, and NIGMS P20GM130459 to SE; NHLBI R35HL150778 to MT; NHLBI K01HL138215 to ALG). The Transgenic Genotyping and Phenotyping Core at the COBRE Center for Molecular and Cellular Signaling in the Cardiovascular System, University of Nevada, Reno, is maintained by a grant from NIH/NIGMS (P20GM130459 Sub#5451). The High Spatial and Temporal Resolution Imaging Core at the COBRE Center for Molecular and Cellular Signaling in the Cardiovascular System, University of Nevada, Reno, is maintained by a grant from NIH/NIGMS (P20GM130459 Sub#5452).

## Additional information

### Competing interests

Mohamed Trebak: Reviewing editor, *eLife*. The other authors declare that no competing interests exist.

### Funding

| Funder | Grant reference number | Author |
| --- | --- | --- |
| National Heart, Lung, and Blood Institute | R35HL155008 | Scott Earley |
| National Heart, Lung, and Blood Institute | R01HL137852 | Scott Earley |
| National Heart, Lung, and Blood Institute | R01HL091905 | Scott Earley |
| National Heart, Lung, and Blood Institute | R01HL139585 | Scott Earley |
| National Heart, Lung, and Blood Institute | R01HL122770 | Scott Earley |
| National Heart, Lung, and Blood Institute | R01HL146054 | Scott Earley |
| National Institute of Neurological Disorders and Stroke | RF1NS110044 | Scott Earley |
| National Institute of Neurological Disorders and Stroke | R61NS115132 | Scott Earley |
| National Institute of General Medical Sciences | P20GM130459 | Scott Earley |
| National Heart, Lung, and Blood Institute | R35HL150778 | Mohamed Trebak |
| National Heart, Lung, and Blood Institute | K01HL138215 | Albert L Gonzales |

The funders had no role in study design, data collection and interpretation, or the decision to submit the work for publication.

## Author contributions

Vivek Krishnan, Formal analysis, Investigation, Methodology, Project administration, Writing - original draft, Writing - review and editing; Sher Ali, Formal analysis, Investigation, Methodology, Writing - original draft; Albert L Gonzales, Conceptualization, Formal analysis, Investigation, Methodology, Writing - original draft, Writing - review and editing; Pratish Thakore, Formal analysis, Investigation, Methodology, Visualization, Writing - review and editing; Caoimhin S Griffin, Formal analysis, Investigation, Methodology; Evan Yamasaki, Investigation, Methodology; Michael G Alvarado, Formal analysis, Methodology, Visualization; Martin T Johnson, Methodology, Resources; Mohamed Trebak, Conceptualization, Funding acquisition, Resources, Supervision, Writing - original draft, Writing - review and editing; Scott Earley, Conceptualization, Data curation, Formal analysis, Funding acquisition, Investigation, Methodology, Project administration, Resources, Supervision, Validation, Writing - original draft, Writing - review and editing

## Author ORCIDs

Vivek Krishnan (ID) http://orcid.org/0000-0001-5064-4910
Pratish Thakore (ID) http://orcid.org/0000-0002-2086-5453
Michael G Alvarado (ID) http://orcid.org/0000-0002-3489-9021
Mohamed Trebak (ID) http://orcid.org/0000-0001-6759-864X
Scott Earley (ID) http://orcid.org/0000-0001-9560-2941

## Ethics

All animal studies were performed in strict accordance with the guidelines of the Institutional Animal Care and Use Committee (IACUC) of the University of Nevada, Reno, and in accordance with the approved protocol 20-06-2020. All surgery was performed under isoflurane anesthesia, and every effort was made to minimize suffering, including preoperative analgesia provided by subcutaneous injection of 50 µ ;g/kg buprenorphine.

## Decision letter and Author response

Decision letter https://doi.org/10.7554/eLife.70278.sa1
Author response https://doi.org/10.7554/eLife.70278.sa2

# Additional files

## Supplementary files

• Transparent reporting form

• Supplementary file 1. Percentage of overlapping clusters for BK:RyR2, RyR2:BK, TRPM4:IP3R, and IP3R:TRPM4 in control and Stim1-smKO cells.

• Supplementary file 2. Forward and reverse primer sequences used for ddPCR experiments.

## Data availability

All data generated or analyzed during this study are included in the manuscript and supporting files. All source data files and blots images have been provided.

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

# Appendix 1

## Appendix 1—key resources table

| Reagent type (species) or resource | Designation | Source or reference | Identifiers | Additional information |
|---|---|---|---|---|
| Genetic reagent (*Mus musculus*) | C57BL/6J; wild type | Jackson Laboratory | Strain# 000664; RRID:IMSR_JAX:000664 | |
| Genetic reagent (*M. musculus*) | Stim1$^{fl/fl}$ | Jackson Laboratory | Strain# 023350; RRID:IMSR_JAX:023350 | |
| Genetic reagent (*M. musculus*) | Myh11$^{Cre}$; Myh11$^{Cre}$/ERT2 | Jackson Laboratory | Strain# 019079; RRID:IMSR_JAX:019079 | |
| Antibody | Anti-Stim1 (rabbit polyclonal) | Sigma-Aldrich | Cat# S6072; RRID:AB_1079008 | (1:1000) |
| Antibody | Anti-Stim1 (rabbit polyclonal) | Cell Signaling Technologies | Cat# 4916; RRID:AB_2271287 | (1:100) |
| Antibody | Anti-Stim1 (mouse monoclonal) | BD Biosciences | Cat# 610954; RRID:AB_398267 | (1:50) |
| Antibody | Anti-BKα1 (rabbit polyclonal) | Alomone Labs | Cat# APC-021; RRID:AB_2313725 | (1:100) |
| Antibody | Anti-RyR2 (mouse monoclonal) | Thermo Fisher Scientific | Cat# MA3-916; RRID:AB_2183054 | (1:50) |
| Antibody | Anti-TRPM4 (goat polyclonal) | https://Antibodies-online.com | Cat# ABIN572220; RRID:AB_10787216 | (1:400) |
| Antibody | Anti-IP$_3$R (rabbit polyclonal) | Abcam | Cat# ab5804; RRID:AB_305124 | (1:200) |
| Antibody | Alexa Fluor 647-conjugated anti-mouse IgG (goat polyclonal) | Thermo Fisher Scientific | Cat# A-21236; RRID:AB_2535805 | (1:1000) |
| Antibody | Alexa Fluor 532-conjugated anti-rabbit IgG (goat polyclonal) | Thermo Fisher Scientific | Cat# A-11009; RRID:AB_2534076 | (1:1000) |
| Antibody | Alexa Fluor 647-conjugated anti-goat IgG (donkey polyclonal) | Abcam | Cat# ab150131; RRID:AB_2732857 | (1:1000) |
| Antibody | Alexa Fluor 568-conjugated anti-rabbit IgG (donkey polyclonal) | Thermo Fisher Scientific | Cat# A10042; RRID:AB_2534017 | (1:1000) |
| Other | ER-Tracker Green | Thermo Fisher Scientific | Cat# E34251 | (1:1000) |
| Other | Cell-Mask Deep Red | Thermo Fisher Scientific | Cat# C10046 | (1:1000) |
| Other | Fluo-4 AM | Thermo Fisher Scientific | Cat# F14201 | 1 µM |
| Other | Fura-2 AM | Thermo Fisher Scientific | Cat# F1221 | 4 µM |
| Chemical compound, drug | Tamoxifen | Sigma-Aldrich | Cat# T5648 | |
| Chemical compound, drug | Triton X-100 | Sigma-Aldrich | CAS# 9036-19-5 | |
| Chemical compound, drug | SEABLOCK | Thermo Fisher Scientific | Cat# 37527 | |
| Chemical compound, drug | Cysteamine hydrochloride | Sigma-Aldrich | CAS# 156-57-0 | |

*Appendix 1 Continued on next page*

*Appendix 1 Continued*

| Reagent type (species) or resource | Designation | Source or reference | Identifiers | Additional information |
|---|---|---|---|---|
| Chemical compound, drug | Twinsil | Picodent | Cat# 1300 5000 | |
| Chemical compound, drug | TRIzol | Thermo Fisher Scientific | Cat# 15596026 | |
| Chemical compound, drug | Synta66 | Sigma-Aldrich | Cat# SML1949 | |
| Chemical compound, drug | Paxilline | Sigma-Aldrich | Cat# P2928 | |
| Chemical compound, drug | 9-Phenanthrol | Sigma-Aldrich | Cat# 211281 | |
| Peptide, recombinant protein | Papain | Worthington Biochemical Corporation | Cat# LS003119 | |
| Peptide, recombinant protein | Collagenase type 2 | Worthington Biochemical Corporation | Cat# LS004202 | |
| Peptide, recombinant protein | Glucose oxidase | Sigma-Aldrich | Cat# G2133 | |
| Peptide, recombinant protein | Catalase | Sigma-Aldrich | Cat# C40 | |
| Commercial assay or kit | 5X RIPA buffer with Protease Inhibitor Cocktail | Cell Biolabs | Cat# AKR-190 | |
| Commercial assay or kit | BCA Protein Assay Kit | Thermo Fisher Scientific | Cat# 23225 | |
| Commercial assay or kit | 12–230 kDa Separation module | ProteinSimple | Cat# SM-W004 | |
| Commercial assay or kit | Anti-Rabbit Detection Module | ProteinSimple | Cat# DM-001 | |
| Commercial assay or kit | Total Protein Detection Module | ProteinSimple | Cat# DM-TP01 | |
| Commercial assay or kit | Direct-zol RNA microprep kit | Zymo Research | Cat# R2060 | |
| Commercial assay or kit | qScript cDNA Supermix | Quanta Biosciences | Cat# 95047 | |
| Commercial assay or kit | QX200 ddPCR EvaGreen Supermix | Bio-Rad | Cat# 186-4033 | |
| Software, algorithm | Compass for SW software | ProteinSimple | https://www.proteinsimple.com/compass/downloads | |
| Software, algorithm | Imaris software | Bitplane/Oxford Instruments | https://imaris.oxinst.com/packages; RRID:SCR_007370 | |
| Software, algorithm | Leica Application Suite X software | Leica | https://www.leica-microsystems.com/products/microscope-software/details/product/leica-las-x-ls/ RRID:SCR_013673 | |
| Software, algorithm | ImageJ software | National Institutes of Health | https://imagej.net/; RRID:SCR_003070 | |
| Software, algorithm | pClamp software | Molecular Devices, LLC | http://www.moleculardevices.com/products/software/pclamp.html; RRID:SCR_011323 | |

*Appendix 1 Continued on next page*

*Appendix 1 Continued*

| Reagent type (species) or resource | Designation | Source or reference | Identifiers | Additional information |
|---|---|---|---|---|
| Software, algorithm | SparkAn custom software | Dr. Adrian Bonev and Dr. Mark Nelson; PMID:22095728 | N/A | Software used to analyze $Ca^{2+}$ spark events kindly provided by Dr. Adrian Bonev and Dr. Mark Nelson from the University of Vermont |
| Software, algorithm | IonWizard software | IonOptix, LLC | https://www.ionoptix.com/products/software/ionwizard-core-and-analysis/ | |
| Software, algorithm | Ponemah software | Data Science International | https://www.datasci.com/products/software/ponemah; RRID:SCR_017107 | |
| Software, algorithm | GraphPad Prism Software | GraphPad Software, Inc | https://www.graphpad.com/; RRID:SCR_002798 | |

