## [Decision Letter]

**Decision letter after peer review:**

Thank you for submitting your article "Peripheral Coupling Sites Formed by STIM1 Govern the Contractility of Vascular Smooth Muscle Cells" for consideration by *eLife*. Your article has been reviewed by 3 peer reviewers, one of whom is a member of our Board of Reviewing Editors, and the evaluation has been overseen by Kenton Swartz as the Senior Editor. The reviewers have opted to remain anonymous.

Essential revisions:

All three reviewers expressed considerable interest in the work. However, they identified important essential revisions that will be needed to support the main conclusions. These revisions could be addressed via a combination of a moderate number of new experiments and text changes:

1. Where is STIM1 clustered relative to other channels? Are the channel clusters measured located at SR-PM junctions. And how does the channel cluster intensity of affected by the STIM1 KO? This very important question can be addressed using imaging approaches already employed in the study.

2. Examine STOCs and TICCs and vasoconstriction using pharmacological tools for BK, TRPM4, Ry2R2 and/or IP3Rs to determine if reduced vasoconstriction in STIM1 knockout arteries is due to modified activity of the same ion channels identified by patch-clamp. This will help link the myography data to the observed changes in channel clusters.

3. How are changes in spark properties linked to alterations in STOCs and TICCs? This is more ambitious but can be addressed by localizing and the quantifying the abundance of CaV, SERCA, PMCA, NCX and by monitoring sparks evoked by depolarization. This would help connect changes in spark generation to the altered currents that are observed.

4. Is the role of STIM1 really independent of Orai and SOCE? Test for effects of CRAC channel blockers (BTP2, CM4620) on sparks, STOCs, TICCs, muscle tone in vitro. Also examine if the localization of STIM1 changes with store depletion.

In addition, there are a numbers of text revisions and clarifications which should be addressed, as described below in the individual reviewer comments.

*Reviewer #1:*

Krishnan and colleagues investigate the role of STIM1 in regulating several aspects of differentiated smooth muscle cell (SMC) function and cell biology relating to the SR-PM junctions and the role of SR-mediated calcium and contractile events. Using isolated smooth muscle cells from the cerebral arteries of control and STIM1 sm-KO mice, the study finds that several key aspects of the anatomy of the plasma membrane-SR contact sites are altered with functional consequences. STIM1 KO SMCs have decreased density and area of PM-SR contact sites, and co-localization of BK channels-RyR2 clusters, TRPM4-IP3R1 protein clusters, and diminished Ca^2+^ spark and STOCs. These defects are manifested in impaired vasoconstriction when challenged with the agonist, U46619, especially following a train of stimulation pulses. The findings on first glance seem to be consistent with a model wherein STIM1 does not necessarily regulate the beat-to-beat contraction of smooth muscle, but instead its role is more apparent in responses to a train of stimulation pulses that requires refilling of ER Ca^2+^ stores and which could be impaired in the absence of STIM1.

Strengths of the manuscript:

The manuscript makes several novel and physiologically relevant observations: the number and size of SR/PM coupling sites is significantly reduced in VSMCs from Stim1-smKO mice, loss of STIM1 alters the microarchitecture of ion channels contained within Ca^2+^-signaling complexes including BK-RyR2 and TRPM4-IP3R complexes, and these changes together alter the properties of Ca^2+^ sparks and reduce BK and TRPM4 channel activity under physiological recording conditions. Consistent with these changes in cell physiology, measurements of contractility reveal that the smooth muscle lacking STIM1 shows impaired contractility and low blood pressure. All together, these are interesting, novel, and physiologically relevant results with potential clinical relevance.

Weaknesses:

The main weakness of the study relates to two conclusions that don't seem to be supported by any compelling data in the study: (i) that STIM1 functions independently of SR Ca^2+^ stores, and (ii) that STIM1 is constitutively active. The authors use these contentions to advance the idea that STIM1 regulates smooth muscle physiology via a mechanism distinct from its role in regulating the well-established SOCE, but this conclusion is needlessly speculative.

– It is really unclear why the authors conclude that the function of STIM1 is independent of its role in regulating SOCE and, moreover, is independent of the SR calcium concentration. These parameters were not measured directly in the present study. In order to support this claim, the study should directly measure the ER Ca to determine whether or not stores are not constitutively depleted. Otherwise, this speculation should be removed.

– The finding that store release is comparable between STIM1-smKO and WT SMCs in response to a single challenge of caffeine (Figure 5I) is insufficient evidence that stores are unaffected by ablating STIM1 expression. What happens to SR calcium stores under conditions of more physiological, repetitive stimulation, for example during a pulse train with caffeine or even U46619? The role of STIM1 may not be apparent following stimulation with a single stimulus, but may become more apparent with challenged with a repetitive pulses that may activate SOCE in order to sustain smooth muscle function during pulsatile contractions.

– The authors put forth the concept that STIM1 is STIM1 is constitutively active. This concept needs further testing, for example by assessing whether STIM1 localization changes following direct store depletion.

Comments for the authors:

1) Examine SOCE in the STIM1 KO cells generated within the study. This is a simple and essential control necessary to assess SOCE levels in differentiated SMCs, and if detected, confirm that the STIM1 KO mice generated in the current study lose SOCE.

2) Assess SR calcium levels in response to repetitive challenges with a GPCR agonist that will causes repetitive SR Ca^2+^ release mimicking Ca^2+^ release events during pulsatile arterial smooth muscle contractions. Is SR Ca^2+^ store content maintained in the STIM1 KO SMCs under these conditions?

3) Assess the stability of STIM1 clusters in resting and store depleted (thapsigargin) treated cells to determine if STIM1 localization is altered by store depletion as seen in non-excitable cells. The relationship of the STIM1 clusters shown in Figure 1 should also be compared to BK and RyR2 clusters found in Figure 2.

*Reviewer #2:*

Fully differentiated, contractile smooth muscle cells express STIM1, but exhibit only minimal store-operated calcium entry (SOCE), a process that is known to be activated by STIM1. The authors aim to identify physiological functions of STIM1 that are independent of SOCE has been achieved in this manuscript.

The Methods used are broad, integrated and include innovative approaches, such as super-resolution microscopy and inducible, conditional knockout mice. The authors have achieved their aim and the conclusions are, in general, supported by the results. This manuscript will have an impact in the field and should stimulate new research in the fields of calcium signaling and vascular biology.

1. The terminology used for the mice should be modified. According to the text on page 5, "SMC-specific Stim1-knockout (Stim1-smKO) mice" are mice that have not yet received tamoxifen. Mice that have not yet received tamoxifen are Cre-positive Stim1fl/fl mice, not Stim1 knockouts. The same paragraph also denotes Stim1-smKO mice injected with sunflower oil as controls. These mice should not be referred to as Stim1-smKO mice as the Stim1 gene would not have been modified by sunflower oil.

2. Mice used for experimentation are young, at between 6 and 8 weeks of age. What is the reason juvenile mice were used for this study, rather than adults?

3. Figure 1A and Supplementary Figure 1. The mean data show that residual STIM1 is present in tissues of the tamoxifen-injected Cre-positive mice, but no bands can be seen on the Western blots. Can the images be improved so that protein bands can be seen in all knockout lanes? The STIM1 blot from brain looks different in the control and STIM1 knockout and may not be representative. The legend for Supplementary Figure 1A indicates that the corresponding Western blots are total protein. What probe or marker was used to label all proteins? The Methods do not appear to provide this information. Please add a molecular weight marker on Figure 1A. Maybe I am missing something, but it is not clear how STIM1 protein can be between 0.1 and 0.9 of total protein if data are normalized to total protein on a scale of 0 to 1.0. This suggests that STIM1 is up to 90 % of total protein and in some cases, individual data points are more than 1.0 of total protein.

4. How does the size of the fluorescent clusters in the Stim1-smKO cells compare to those in the control cells? Are they similar or different?

5. It would be useful to state what percentage of BK clusters overlap with RyR2 clusters and vice versa in control and Stim1-smKO cells. The same question applies to IP3R and TRPM4 clusters. Please provide these data.

6. If epifluorescence was used to measure cluster colocalization, how do you know if two proteins are colocalized at, or nearby, the surface? Could both proteins be located intracellularly? Please include a discussion of why you think the density of RyR2 clusters and the size of BK, TRPM4 and IP3R clusters are lower, whereas the size of RyR2 clusters is larger in the Stim1-smKO cells. Why would a change in the distance between the SR and plasma membranes in response to STIM1 knockout alter cluster properties in this manner?

7. In Figure 2, why was volume used to calculate PM and SR membranes, rather than surface area? How were PM and SR volumes calculated? What criteria were used to establish that the plasma and SR membranes colocalized? Why are the colocalization sites spherical? Wouldn't you expect that the close apposition of two membranes appear as sheets?

8. It is unusual that the mean amplitude of calcium sparks, when expressed as F/F0, is less than 1 (Figure 5D). If F0 is 1, calcium spark amplitudes should be higher than 1. Figure 5 would benefit from traces showing individual or average traces of calcium sparks that illustrate the amplitude and kinetic differences described in control and Stim1-smKO cells.

9. The authors (page 13) state that the effect of STIM1 knockout on BK and TRPM4 channel activity would produce opposing effects on arterial contractility. Myography is performed to address this question, leading to the observation that STIM1 knockout attenuates vasoconstriction. However, no link to altered BK and TRPM4 channel activity is investigated, which would close the loop introduced by the authors here. For example, experiments could be performed to examine if reduced vasoconstriction in STIM1 knockout arteries is due to attenuated IP3R and TRPM4 activity.

10. What is the resolution of the lightsheet imaging system used for the PM-SR colocalization experiments?

11. Please provide references where it was shown that SERCA, the PMCA and Na+/Ca^2+^ exchangers alter Ca^2+^ spark spread and decay in smooth muscle cells as you write in the Discussion on page 18.

12. Effects of STIM1 knockout are studied, but the cellular location and mechanism of STIM1 in control smooth muscle cells is not shown. The authors could provide evidence, or at least discuss in more detail, where STIM1 is located and how they consider STIM1 maintains close spatial proximity of the plasma and sarcoplasmic reticulum membranes. Is STIM1 located where the two membranes are closely opposed? Is STIM1 nearby RyR, IP3R, BK or TRPM4 channels? Does STIM bind to another PM-located protein? Could that be L-type Ca^2+^ channels, TRP channels or other proteins previously shown to be regulated by STIM1?

*Reviewer #3:*

In this paper, Krishnan et al. examine the functional effects of a targeted knock out of STIM1 in vascular smooth muscle cells (VSMCs). Unlike most cells, differentiated VSMCs express very little store-operated calcium entry (SOCE), but the authors demonstrate that knocking out STIM1, the ER Ca^2+^ sensor for SOCE, has a significant impact on the generation of Ca^2+^ sparks and local BK and TRPM4 channel activation, leading to a loss of myogenic tone and hypotension in intact animals. These results are ascribed to a reduction in the number and size of sarcoplasmic reticulum (SR)- plasma membrane (PM) junctions rather than a loss of SOCE. Thus, this study reveals important functions of STIM1 in VSMCs that may be independent of coupling to Orai1 and activation of SOCE. These results add to a growing list of STIM1 functions beyond SOCE that includes regulation of voltage-gated Ca^2+^ channels, Ca^2+^ ATPases, and adenylate cyclase.

On the whole, the experiments were very carefully done and well presented, and the results are clear and convincing. The breadth of the study is also impressive, ranging from super-resolution imaging of junctions, channel localization and co-clustering in single VSMCs, to single-cell measurements of Ca^2+^ sparks and the transient currents they produce, to measurements of the myogenic response in isolated cerebral and mesenteric arteries, and culminating in hypotension in intact animals. The effects on transient currents through BK (STOCs) and TRPM4 channels (TICCs) could explain the effects on myogenic tone and blood pressure, in line with the current understanding of how local coupling between RyR2 and BK channels, and IP3R and TRPM4 channels control VSMC contractile behavior. These results will be of interest to researchers studying calcium signaling mechanisms and cardiovascular physiology.

The main uncertainty in the paper is mechanistically how knockout of STIM1 causes the observed effects on SR-PM junctions, Ca^2+^ sparks and local activation of BK and TRPM4 channels leading to reduced myogenic response and hypotension. Does STIM1 play a direct role in forming junctions, e.g., through binding of its polybasic domain to negatively charged phospholipids in the PM, or a supporting role where STIM1 binds to microtubules that extend SR tubules towards the PM? It is also unclear whether the observed reduction of coupling sites can explain the altered Ca^2+^ spark characteristics and downstream effects on STOCs and TICCs as proposed by the authors. Changes in channel clustering after STIM1 knockout were not entirely consistent with changes in coupling site density, and it is also possible that STIM1 helps channels to accumulate at junctions. Finally, the possibility that STIM1 may signal through activation of Orai1 and SOCE should be considered. Although differentiated VSMCs express very little STIM1 and Orai1, this study demonstrates that the amount of STIM1 is nevertheless functionally important, and local activation of Orai1 may be able to influence SR Ca^2+^ content or BK or TRPM4 channel activity.

Comments for the authors:

1. The title of the paper states "Peripheral Coupling Sites Formed by STIM1 Govern the Contractility of Vascular Smooth Muscle Cells," but there is no evidence that STIM1 actually forms the sites. How is STIM1 involved? Does STIM1 play a direct role; e.g., a tethering function due to interaction of its polybasic domain with PIP2 in the PM or by promoting SR extension towards the PM by binding to EB1 on the tips of microtubules? Or is it indirect – perhaps reducing the overall amount of SR (Figure 2) reduces the density of SR-PM contacts as a simple mass-action effect. In the absence of any experimental evidence for a direct role, the title and conclusions should be toned down, and these possibilities should be discussed.

2. The effects of STIM1 KO on spark characteristics, BK and TRPM4 activation (STOCs and TICCs), myogenic tone, and blood pressure are clear and convincing. The major question is, how does the loss of STIM1 lead to these effects? The authors propose that the effects arise from a loss of peripheral coupling sites, but the data are not entirely consistent on this point. For example, changes in junction number and size do not always go hand in hand with channel cluster numbers and size. Despite significant reduction of SR-PM coupling site density (Figure 2), there is no change in BK, TRPM4, or IP3R cluster density (Figure 3B, 4B, 4C). Similarly, there is no change in RyR2 cluster size (Figure 3C) even though junctions are smaller. These results do not fit neatly with the idea that changes in junction density and size are responsible for the functional effects on VSMC contractility.

3. STIM1 KO significantly alters spark characteristics, but it is hard to imagine that the increased amplitude, duration, and decay time constant all result simply from a modest decline in the number and volume of SR-PM contacts. Why are they bigger and spread further after STIM1 KO? Also, are the altered spark characteristics consistent with STOC and TICC characteristics? One might expect a lengthening of STOC/TICC events given the increased duration of the sparks.

4. One possibility that should be considered is that some of the effects of STIM1 result from opening Orai1 channels at SR-PM junctions. A small amount of STIM1 and Orai1 were detected in VSMCs by Potier et al. (ref 37). Such activity might be detectable locally but not at a global level, yet could affect local SR Ca^2+^ content or activity of BK or TRPM4 channels. This could be tested by measuring the effects of Orai1 inhibitors on constitutive Ca^2+^ influx, sparks, and STOCs and TICCs.

5. Another possibility is that STIM1 influences the accumulation of channels at SR-PM junctions. Do clusters of STIM1 colocalize with clusters of BK and TRPM4? Does STIM1 KO affect the number of channels at junctions? This will contribute to the size of the local STOC and TICC signals.

6. It is unclear what proportion of the detected channel clusters are actually located at junctions. GSDIM was performed in epifluorescence mode, which implies that signals were collected throughout the cell thickness, not necessarily near the PM. Cluster densities for the channels differ: BK (15/µm^2^), RyR2 (4-6/µm^2^), TRPM4 (40/µm^2^), and IP3R clusters (30/µm^2^) and are much higher than the density of colocalized clusters (BK-RyR2 are 0.3/µm^2^, and IP3R-TRPM4 are 3/µm^2^). It seems likely that most of these clusters are actually not associated with junctions. If the clusters are not at junctions, then the quantification of cluster density cannot be related to the frequency of STOCs or TICCs.

[Editors' note: further revisions were suggested prior to acceptance, as described below.]

Thank you for resubmitting your work entitled "Peripheral Coupling Sites Formed by STIM1 Govern the Contractility of Vascular Smooth Muscle Cells" for further consideration by *eLife*. Your revised article has been reviewed by 3 peer reviewers, one of whom is a member of our Board of Reviewing Editors, and the evaluation has been overseen by Kenton Swartz as the Senior Editor, and a Reviewing Editor.

The manuscript has been improved but there are some remaining issues that need to be addressed, as outlined below:

(1) The manuscript does not actually show that STIM1 is present at the SR-PM junctions that are the focus of the paper. Rather, the data in Figure 2 only shows that there is an overall decrease in the number of SR-PM junctions in the STIM1 KO. In the absence of direct proof that STIM1 is located at the junctions, the idea that STIM1 drives the formation of the SR-PM contacts is speculative. Therefore, the title and the discussion of the paper should reflect this point as indicated by reviewers 1 and 3.

(2) This point was previously raised. The fraction of BK (and TRPM4 channels) that are actually co-localized with STIM1 is really quite small to begin with. This is readily apparent in the co-localization images of Figure 5 – most BK and TRPM4 channels don't cluster with STIM1. Authors must explain explicitly the relationship between channel clusters and coupling sites in the text. See also comments by reviewers 1 and 3 on this issue.

(3) Figure 2 clearly shows that the volume of the SR is strongly diminished in the STIM1 KO. It seems natural to expect then that as the volume of the SR declines and it pulls away from the membrane, SR-PM contacts will also go down. This issue was raised in the previous review. Does this decrease in SR volume alone not account for the observed decline in SR-PM contacts? This question needs to be explicitly addressed in the manuscript with a well-rounded discussion of what the data actually show.

*Reviewer #1:*

Reviewers have addressed my previous comments well and the paper is much improved.

I do have two further points that would need addressing:

– First, nowhere in the paper is there data showing that STIM1 is actually present at the peripheral ER-PM sites. I agree that the data showing that STIM1 KO decreases interactions between the PM and SR are in VSMCs is strong and convincing. But the paper strongly implies that STIM1 forms ("fosters") these contacts. Can the authors show that STIM1 labelling is present at the SR-PM contact sites? In the absence of this data, the conclusion that STIM1 is crucial for "fostering SR-PM coupling" is indirect and in the worst case, could be entirely incidental.

– Figure 5: The co-localization of STIM1 with BK (and TRPM4) is actually pretty hard to see and less than compelling. Arrows should be used on the BK and TRPM4 panels to precisely denote which BK clusters are co-localized with which STIM1 clusters. I also don't understand the statements indicating that "BK and TRPM4 colocalize with STIM1 more frequently than would be predicted if the proteins are randomly distributed". How was the random distribution modelled? How many channels? What is the area of the membrane? How was cluster formation modelled/induced in the random distribution model? With sufficient numbers of BK and STIM1 proteins, one would expect that random distribution may even cause co-localization of the observed numbers of BK and STIM1 proteins. These important details that are crucial to understanding whether co-localization is mechanistically driven are missing.

*Reviewer #2:*

The manuscript has been significantly improved.

*Reviewer #3:*

The authors have done a good job of addressing the comments and the manuscript has been improved. There are only a few remaining concerns that I would like the authors to address.

1. As for the question of whether channel clusters are present at SR-PM junctions, the authors should explain more clearly in the text what fraction of the clusters are likely to reside at these sites. See my comment on point #2, below.

4. Good – looks like there really isn't any contribution from SOCE. In Figure 1—figure supplement 3E, please state how the comparative SOCE response was measured – peak 340/380 ratio, or the slopes from the data in D?

My review:

1. My comment on the title did not dispute the effect of STIM1 KO on the number of peripheral coupling sites. The problem is that the wording suggests that STIM1 plays a direct role in forming these sites, and that is still to my knowledge an unanswered question in the field. Elsewhere in the paper the authors state that STIM1 maintains coupling, or is important for stable coupling, which is accurate. However, the title implies STIM1 acts in the formation of junctions, and without more mechanistic studies this is still an open question. I would suggest rewording the title to make it more accurate – perhaps referring to these sites as STIM1-dependent rather than STIM1-formed.

2. My comment relates to the lack of correspondence between channel clusters and coupling sites. As I stated, STIM1 KO greatly reduces coupling site density, but does not change BK, TRPM4 or IP3R cluster density. In addition, coupling sites are 0.17/µm2 in WT, but clusters are much more prevalent: 16/µm2 for BK, 40/µm2 for TRPM4, 32 for IP3R. This 100x difference indicates that many of the channel clusters are not associated with an identified coupling site, and probably for this reason the number is not grossly affected by loss of coupling sites. This conclusion is supported by the data of Figure 5, where only a small fraction of BK and TRPM4 channels colocalize with STIM1 puncta (indicating SR-PM junctions). It would be helpful to explain explicitly the relationship between channel clusters and coupling sites in the text. I may be missing something, but I do not understand how stochastic self-assembly in the rebuttal explains how loss of STIM1 affects cluster size. It is hard for me to imagine that given the very small fraction of channel clusters associated with SR-PM junctions (~5%) in WT cells, how the observed reduction in SR-PM coupling sites could lead to a measurable cell-wide decrease in the size of clusters as the authors suggest.

---

## [Author Response]

Essential revisions:All three reviewers expressed considerable interest in the work. However, they identified important essential revisions that will be needed to support the main conclusions. These revisions could be addressed via a combination of a moderate number of new experiments and text changes:1. Where is STIM1 clustered relative to other channels? Are the channel clusters measured located at SR-PM junctions. And how does the channel cluster intensity of affected by the STIM1 KO? This very important question can be addressed using imaging approaches already employed in the study.

We performed several new experiments to address this set of concerns. We utilized the TIRF modality of our GSDIM system to image STIM1 protein clusters in VSMCs. This method detects protein clusters at or near the plasma membrane, to a depth of ~ 150 nm. These data show that STIM1 protein clusters are present at the plasma membrane, and that these clusters were significantly reduced in size and density in VSMCs from *Stim1-*smKO mice (Figure 1—figure supplement 2). We also imaged VSMCs coimmunolabeled with BK and STIM1 or TRPM4 and STIM1 using TIRF-mode GSDIM, and analyzed the data using object-based analysis (OBA) (Figure 5). These data show that BK and TRPM4 colocalize with STIM1 more frequently than would be predicted if the proteins are randomly distributed, providing evidence that the channels form nanoscale complexes with STIM1 at or near the plasma membrane.

Our data also show that STIM1 knockout does not change the density of BK and TRPM4 channel clusters (Figures 3B and 4B). However, STIM1 knockout is associated with reduced size of BK and TRPM4 channel clusters (Figures 3B and 4B), suggesting that STIM1 may be involved with the formation of protein clusters on the plasma membrane. Stochastic self-assembly is a currently accepted model of protein cluster formation at the plasma membrane (1). Briefly, data from this paper show that protein clusters within the plasma membrane grow in size over time, and the probability of removal of protein clusters from the membrane also increases over time. The net result is an exponential distribution of protein cluster sizes, matching the findings reported by multiple laboratories. In this context, a reasonable interpretation of our data is that when STIM1 is present, the removal of BK and TRPM4 channel clusters from the membrane is slower, allowing them to grow to larger sizes. Knockout of STIM1 removes this protective effect and BK and TRPM4 channels are removed more quickly, at a smaller size. We envision that coupling between the SR and plasma membrane slows the removal of protein clusters from the membrane.

2. Examine STOCs and TICCs and vasoconstriction using pharmacological tools for BK, TRPM4, Ry2R2 and/or IP3Rs to determine if reduced vasoconstriction in STIM1 knockout arteries is due to modified activity of the same ion channels identified by patch-clamp. This will help link the myography data to the observed changes in channel clusters.

We performed the suggested experiments and have added new data showing the effects of the BK channel inhibitor paxilline and the TRPM4 channel inhibitor 9-phenanthrol on vasoconstriction of cerebral arteries isolated from control and *Stim1-*smKO mice (Figure 8—figure supplement 1). As expected, we observed that myogenic tone increased in response to paxilline treatment in cerebral arteries isolated from control mice. Cerebral arteries from *Stim1-*smKO mice have little basal myogenic tone and showed only very slight increase in myogenic tone with paxilline treatment, indicating that these vessels have very little BK channel activity. As we have previously reported (2), 9-phenathrol treatment resulted in the loss of myogenic tone of cerebral arteries from control mice (Figure 8—figure supplement 1). Cerebral arteries from *Stim1-*smKO mice have very little basal tone and exhibited only a slight dilation in response to 9-phenathrol treatment, suggesting that these vessels have very little TRPM4 activity. Together, these data link the patch-clamp and pressure myography data.

3. How are changes in spark properties linked to alterations in STOCs and TICCs ? This is more ambitious but can be addressed by localizing and the quantifying the abundance of CaV, SERCA, PMCA, NCX and by monitoring sparks evoked by depolarization. This would help connect changes in spark generation to the altered currents that are observed.

The experiments proposed by the reviewers to localize and quantify the abundance of CaV, SERCA, PMCA, NCX in control and STIM1 knockout mice are fascinating, but would require a massive effort over several months and greatly increase the length of this already robust manuscript. We plan to investigate these questions in a follow-up study.

The changes in Ca^2+^ spark properties associated with STIM1 knockout are not the primary cause of the loss of STOC and TICC frequency. Our model indicates that STOC and TICC activity is diminished by the knockout of STIM1 because the functional coupling of BK and TRPM4 channels on the plasma membrane and RyR2s and IP_3_Rs on the SR is lost. Functional coupling is lost because peripheral coupling between the membranes is disrupted.

We propose that disruption of peripheral coupling between the SR and PM is also responsible for altering the properties of Ca^2+^ sparks. Ca^2+^ sparks are produced within microdomains that are formed by the close association of the SR and plasma membrane. Separation of the two membranes enlarges the area of these Ca^2+^ signaling microdomains, leading to the observed increase in spatial spread. Enlargement of the microdomains also increases the distance between the source of the Ca^2+^ spark and the SERCA and PMCA pumps and Na/Ca^2+^ exchangers which remove Ca^2+^ from the cytosol (3-5), leading to prolonged decay and increased amplitude. We have attempted to clarify these concepts in the revised manuscript.

4. Is the role of STIM1 really independent of Orai and SOCE? Test for effects of CRAC channel blockers (BTP2, CM4620) on sparks, STOCs, TICCs, muscle tone in vitro. Also examine if the localization of STIM1 changes with store depletion.

Several studies in various vascular beds showed that SOCE and CRAC channel activity are undetectable in contractile VSMCs from systemic arteries. Nevertheless, we used the more selective Orai1 channel blocker Synta66 to address the reviewer’s question. Synta66 had no effect on STOCs or TICCs in patch-clamp clamp experiments (Figure 7—figure supplements 1 and 2) or on myogenic tone in pressure myography experiments (Figure 8—figure supplement 1). We also imaged STIM1 clusters in cerebral artery SMCs after thapsigargin treatment using GSDIM-TIRF and found no increase in STIM1 cluster density or size compared to vehicle-treated cells (Figure 1—figure supplement 3). In addition, in agreement with prior studies (6, 7), we provide new data showing that only trivial levels of SOCE can be evoked in native, contractile SMCs from cerebral arteries when compared with proliferative cerebral artery SMCs (Figure 1—figure supplement 3). Together, these data demonstrate that STIM1's function in contractile SMCs is independent of Orai1 and SOCE.

In addition, there are a numbers of text revisions and clarifications which should be addressed, as described below in the individual reviewer comments.Reviewer #1:[…] Comments for the authors:1) Examine SOCE in the STIM1 KO cells generated within the study. This is a simple and essential control necessary to assess SOCE levels in differentiated SMCs, and if detected, confirm that the STIM1 KO mice generated in the current study lose SOCE.

We performed SOCE experiments in proliferative and fully differentiated contractile cerebral artery SMCs. In agreement with prior studies (6, 7), we found that proliferative SMCs exhibited robust SOCE whereas contractile SMCs had almost none (Figure 1 —figure supplement 3).

2) Assess SR calcium levels in response to repetitive challenges with a GPCR agonist that will causes repetitive SR Ca^2+^ release mimicking Ca^2+^ release events during pulsatile arterial smooth muscle contractions. Is SR Ca^2+^ store content maintained in the STIM1 KO SMCs under these conditions?

We attempted to assess SR Ca^2+^ levels in response to repeated applications of U46619 – a thromboxane A2 receptor vasoconstrictor agonist – but the native contractile SMCs used for these studies maximally contracted in response to this treatment, and were very slow to relax. This generated imaging artifacts, rendering the experiments infeasible. To address the reviewer's concerns, we provide references to a prior study showing that STIM1 knockdown/knockout does not significantly alter resting ER Ca^2+^ levels (8-10). Thus, our finding that STIM1 knockout does not alter SR Ca^2+^ levels is in agreement with the prior literature.

3) Assess the stability of STIM1 clusters in resting and store depleted (thapsigargin) treated cells to determine if STIM1 localization is altered by store depletion as seen in non-excitable cells. The relationship of the STIM1 clusters shown in Figure 1 should also be compared to BK and RyR2 clusters found in Figure 2.

We performed these experiments and found that depletion of SR Ca^2+^ stores using thapsigargin did not change STIM1 protein cluster size or density (Figure 1—figure supplement 3). In addition, we imaged co-immunolabeled VSMCs using TIRF-mode GSDIM and found that STIM1 colocalized with BK and TRPM4 channels at the plasma membrane more frequently than would be predicted by random distribution, suggesting that STIM1 forms nanoscale complexes with these channels (Figure 5).

Reviewer #2:[…] 1. The terminology used for the mice should be modified. According to the text on page 5, "SMC-specific Stim1-knockout (Stim1-smKO) mice" are mice that have not yet received tamoxifen. Mice that have not yet received tamoxifen are Cre-positive Stim1fl/fl mice, not Stim1 knockouts. The same paragraph also denotes Stim1-smKO mice injected with sunflower oil as controls. These mice should not be referred to as Stim1-smKO mice as the Stim1 gene would not have been modified by sunflower oil.

We agree, and have changed the description to clarify that *Myh11^Cre^*-*Stim1^fl/fl^* mice injected with tamoxifen are referred to as *Stim1*-smKO mice and *Myh11^Cre^*-*Stim1^fl/fl^* mice injected with vehicle (sunflower oil) are referred to as control mice.

2. Mice used for experimentation are young, at between 6 and 8 weeks of age. What is the reason juvenile mice were used for this study, rather than adults?

Based on existing guidelines and studies using tamoxifen-induced activation of *Cre* recombinase in smooth muscle (11-14), the optimum age for tamoxifen injection to induce *Cre* recombinase activity via *i.p.* injections at a dose which minimizes other side effects is between 4 to 6 weeks old. The maximum *Cre* recombinase activity is then observed to be at 7 days after the final injection, which is why the mice are between 6 and 8 weeks old when we use them for experimentation.

3. Figure 1A and Supplementary Figure 1. The mean data show that residual STIM1 is present in tissues of the tamoxifen-injected Cre-positive mice, but no bands can be seen on the Western blots. Can the images be improved so that protein bands can be seen in all knockout lanes? The STIM1 blot from brain looks different in the control and STIM1 knockout and may not be representative. The legend for Supplementary Figure 1A indicates that the corresponding Western blots are total protein. What probe or marker was used to label all proteins? The Methods do not appear to provide this information. Please add a molecular weight marker on Figure 1A. Maybe I am missing something, but it is not clear how STIM1 protein can be between 0.1 and 0.9 of total protein if data are normalized to total protein on a scale of 0 to 1.0. This suggests that STIM1 is up to 90 % of total protein and in some cases, individual data points are more than 1.0 of total protein.

Please note, as stated in the methods section, we used Wes capillary electrophoresis and not Western blot for immunodetection of STIM1. The images shown in our manuscript are a graphical representation of the chromatograms produced by the Wes system. This is currently the standard way of presenting the data. We have improved the quality of these images to show faint STIM1 bands in the knockout lanes and added a better representative blot for the brain samples. We have also added a molecular weight marker to Figure 1A. For labeling total protein, we use the Total Protein Detection Module *for Wes* from Proteinsimple. It utilizes biotin labeling of all proteins in the lanes which are then detected using streptavidin-HRP chemiluminescence, whereas the STIM1 band is detected by immunolabeling with Anti-Stim1 antibody. In other words, the ratio of STIM1 band density to total protein band density does not indicate the fraction of STIM1 that makes up total protein. We have added this information to the methods section.

4. How does the size of the fluorescent clusters in the Stim1-smKO cells compare to those in the control cells? Are they similar or different?

The size and density of the STIM1 fluorescent clusters is significantly smaller in VSMCs from *Stim1-*smKO mice compared with controls (Figure 1 and Figure 1—figure supplement 2).

5. It would be useful to state what percentage of BK clusters overlap with RyR2 clusters and vice versa in control and Stim1-smKO cells. The same question applies to IP3R and TRPM4 clusters. Please provide these data.

The data requested are shown in Table S1.

6. If epifluorescence was used to measure cluster colocalization, how do you know if two proteins are colocalized at, or nearby, the surface? Could both proteins be located intracellularly? Please include a discussion of why you think the density of RyR2 clusters and the size of BK, TRPM4 and IP3R clusters are lower, whereas the size of RyR2 clusters is larger in the Stim1-smKO cells. Why would a change in the distance between the SR and plasma membranes in response to STIM1 knockout alter cluster properties in this manner?

We used the epifluorescence mode for our initial superresolution imaging experiments because we wanted to detect membrane proteins (BK and TRPM4) and intracellular proteins (RyR2, IP3Rs, and STIM1). This approach has been supplemented with TIRF-mode images in the revised manuscript (Figure 1—figure supplement 2, Figure 5).

Effects of STIM1 knockout on protein cluster sizes – Stochastic self-assembly is a currently accepted model of protein cluster formation at the plasma membrane (1). Briefly, data from this paper show that protein clusters within the plasma membrane grow in size over time, and the probability of removal of protein clusters from the membrane also increases over time. The net result is an exponential distribution of protein cluster sizes, matching the findings reported by multiple laboratories. In this context, a reasonable interpretation of our data is that when STIM1 is present, the removal of BK and TRPM4 channel clusters from the membrane is slower, allowing them to grow to larger sizes. Knockout of STIM1 removes this protective effect and BK and TRPM4 channels are removed more quickly, at a smaller size. We envision that coupling between the SR and plasma membrane slows the removal of protein clusters from the membrane. As for IP_3_R and RyR2 clusters, very little is known about protein cluster formation in the SR membrane. We expect that a conceptual framework for this process will emerge in the near future.

7. In Figure 2, why was volume used to calculate PM and SR membranes, rather than surface area? How were PM and SR volumes calculated? What criteria were used to establish that the plasma and SR membranes colocalized? Why are the colocalization sites spherical? Wouldn't you expect that the close apposition of two membranes appear as sheets?

We collected 3D images of the SR and PM of SMCs using the SIM modality of our lattice lightsheet microscope (Figure 2). Co-localization was defined as areas where SR and PM voxels (i.e., pixels in 3D) exist in the same space at the resolution limit of our system. The lateral resolution limit for these experiments is on the order of 300 nm and the smallest volume of interaction that can be resolved under these conditions is approximately 300 x 300 x 300 nm or 0.027 μm^3^. We are not aware of any other study that has investigated the shape of SR/PM interactions in 3D throughout entire native SMCs cells at this level of resolution and we did not know what to expect. The data show that SR/PM colocalizing sites formed well-defined 3D structures that were either irregular ellipsoid or roughly spherical. We therefore reported the volume of these 3D structures. We agree that in 2D representations, the two membranes would appear as sheets (as is the case for TEM experiments) and interactions would properly be quantified as area rather than volume.

8. It is unusual that the mean amplitude of calcium sparks, when expressed as F/F0, is less than 1 (Figure 5D). If F0 is 1, calcium spark amplitudes should be higher than 1. Figure 5 would benefit from traces showing individual or average traces of calcium sparks that illustrate the amplitude and kinetic differences described in control and Stim1-smKO cells.

The amplitudes of Ca^2+^ sparks are expressed as ΔF/F0 rather than F/F0. Therefore, amplitudes are less than 1. Representative traces have been added to Figure 6 of the revised manuscript.

9. The authors (page 13) state that the effect of STIM1 knockout on BK and TRPM4 channel activity would produce opposing effects on arterial contractility. Myography is performed to address this question, leading to the observation that STIM1 knockout attenuates vasoconstriction. However, no link to altered BK and TRPM4 channel activity is investigated, which would close the loop introduced by the authors here. For example, experiments could be performed to examine if reduced vasoconstriction in STIM1 knockout arteries is due to attenuated IP3R and TRPM4 activity.

To address the reviewer's question, we investigated the effect of pharmacological inhibitors of BK and TRPM4 on contractility of arteries isolated from *Stim1*-smKO mice. Although the basal tone was reduced significantly in the arteries from knockout mice, the BK inhibitor paxilline induced vasodilation and the TRPM4 inhibitor 9-phenanthrol induced vasoconstriction, indicating that the remaining channels which are still coupled with RyR2 and IP_3_R retain their function (Figure 8—figure supplement 1).

10. What is the resolution of the lightsheet imaging system used for the PM-SR colocalization experiments?

Using fluorescent beads, we determined that for the SIM modality of the LLS the resolution for 642 nm wavelength (used for PM labeling) is 250 to 335 nm and the resolution for 488 nm wavelength (used for SR labeling), is 225 to 295 nm.

11. Please provide references where it was shown that SERCA, the PMCA and Na+/Ca^2+^ exchangers alter Ca^2+^ spark spread and decay in smooth muscle cells as you write in the Discussion on page 18.

To clarify, we propose that disruption of peripheral coupling between the SR and PM is also responsible for altering the properties of Ca^2+^ sparks. SERCA, PMCA and Na^+^/Ca^2+^ exchangers are all well-known ca^2+^ sinks which help maintain Ca^2+^ levels intracellularly (3-5). We think the disruption of peripheral coupling between the SR and PM and the Ca^2+^ signaling microdomains within these coupling sites leads to a loss of proximity between the Ca^2+^ spark sources and the Ca^2+^ sinks leading to altered Ca^2+^ spark spread and decay.

12. Effects of STIM1 knockout are studied, but the cellular location and mechanism of STIM1 in control smooth muscle cells is not shown. The authors could provide evidence, or at least discuss in more detail, where STIM1 is located and how they consider STIM1 maintains close spatial proximity of the plasma and sarcoplasmic reticulum membranes. Is STIM1 located where the two membranes are closely opposed? Is STIM1 nearby RyR, IP3R, BK or TRPM4 channels? Does STIM bind to another PM-located protein? Could that be L-type Ca^2+^ channels, TRP channels or other proteins previously shown to be regulated by STIM1?

New data were added to the revised manuscript to address these questions. Using the TIRF modality of our GSDIM system, we detected STIM1 clusters at or near the cell membrane (penetration ~150 nm) in in VSMCs from control animals (Figure 1—figure supplement 2). We also used GSDIM-TIRF to show that STIM1 colocalizes with BK and TRPM4 protein clusters at or near the plasma membrane in cells from control mice (Figure 5). Previous studies have also shown that STIM1 interacts with RyR2 (15), IP_3_R (16) and directly with the plasma membrane via cholesterol binding domains (17). Thus, STIM1 likely interacts with multiple proteins found at peripheral coupling sites, and directly with the plasma membrane. Future studies will explore these mechanisms in detail.

Reviewer #3:[…] Comments for the authors:1. The title of the paper states "Peripheral Coupling Sites Formed by STIM1 Govern the Contractility of Vascular Smooth Muscle Cells," but there is no evidence that STIM1 actually forms the sites. How is STIM1 involved? Does STIM1 play a direct role; e.g., a tethering function due to interaction of its polybasic domain with PIP2 in the PM or by promoting SR extension towards the PM by binding to EB1 on the tips of microtubules? Or is it indirect – perhaps reducing the overall amount of SR (Figure 2) reduces the density of SR-PM contacts as a simple mass-action effect. In the absence of any experimental evidence for a direct role, the title and conclusions should be toned down, and these possibilities should be discussed.

The canonical function of STIM1 is the formation of peripheral coupling sites during store depletion, and a wealth of evidence supports this concept. Our data clearly show that knocking out STIM1 in SMCs reduces the area of interactions between the SR and PM (Figure 2). We define such interactions as "peripheral coupling sites". Therefore, our data show that STIM1 expression is needed to maintain normal levels of peripheral coupling in native contractile SMCs. Further, our data also show that knockout of STIM1 reduces the contractility of isolated cerebral arteries. In other words, our data show a direct link between STIM1 expression, peripheral coupling, and SMC contractility. Accordingly, while we agree with the reviewer that additional studies are required to determine exactly how STIM1 regulates these coupling sites, we contend that our title accurately describes the major findings and conclusions of the manuscript. The detailed biochemical and molecular dissection of STIM1 function suggested by the reviewer is a very exciting approach but is likely not feasible using native (non-cultured) VSMCs and would greatly expand the scope of the current study. We hope to pursue these questions in the future.

2. The effects of STIM1 KO on spark characteristics, BK and TRPM4 activation (STOCs and TICCs), myogenic tone, and blood pressure are clear and convincing. The major question is, how does the loss of STIM1 lead to these effects? The authors propose that the effects arise from a loss of peripheral coupling sites, but the data are not entirely consistent on this point. For example, changes in junction number and size do not always go hand in hand with channel cluster numbers and size. Despite significant reduction of SR-PM coupling site density (Figure 2), there is no change in BK, TRPM4, or IP3R cluster density (Figure 3B, 4B, 4C). Similarly, there is no change in RyR2 cluster size (Figure 3C) even though junctions are smaller. These results do not fit neatly with the idea that changes in junction density and size are responsible for the functional effects on VSMC contractility.

Our data show that STIM1 knockout reduces the size of BK and TRPM4 protein cluster size at the plasma membrane. Stochastic self-assembly is a currently accepted model of protein cluster formation at the plasma membrane (1). Briefly, this model states that protein clusters within the plasma membrane grow in size over time, and the probability of removal of these clusters from the membrane also increases over time. The net result is an exponential distribution of protein cluster size, matching the findings reported by multiple laboratories. Applying this model, a reasonable interpretation of our data is that when STIM1 is present, the removal of BK and TRPM4 channel clusters from the membrane is slower, allowing them to grow to larger sizes. Knockout of STIM1 removes this protective effect and BK and TRPM4 channels are removed more quickly, at a smaller size. We envision that coupling between the SR and plasma membrane slows the removal from the membrane, but this needs experimental verification. As for IP_3_R and RyR2 clusters, very little is known about protein cluster formation in the SR membrane. We expect that a conceptual framework will emerge in the future.

3. STIM1 KO significantly alters spark characteristics, but it is hard to imagine that the increased amplitude, duration, and decay time constant all result simply from a modest decline in the number and volume of SR-PM contacts. Why are they bigger and spread further after STIM1 KO? Also, are the altered spark characteristics consistent with STOC and TICC characteristics? One might expect a lengthening of STOC/TICC events given the increased duration of the sparks.

Our interpretation of these data is that disruption of peripheral coupling between the SR and PM enlarges the area of Ca^2+^ signaling microdomains, leading to the observed increase in spatial spread. Enlargement of the microdomains increases the distance between source of the Ca^2+^ spark and SERCA, PMCA, and Na^+^/Ca^2+^ exchangers which remove Ca^2+^ from the cytosol (3-5), leading to prolonged decay. We observed a reduction in STOC frequency as well as amplitude despite the increase in Ca^2+^ spark amplitude, duration and decay time which further supports our idea that there is a loss of functional coupling between RyR2 and BK channels in the absence of STIM1.

4. One possibility that should be considered is that some of the effects of STIM1 result from opening Orai1 channels at SR-PM junctions. A small amount of STIM1 and Orai1 were detected in VSMCs by Potier et al. (ref 37). Such activity might be detectable locally but not at a global level, yet could affect local SR Ca^2+^ content or activity of BK or TRPM4 channels. This could be tested by measuring the effects of Orai1 inhibitors on constitutive Ca^2+^ influx, sparks, and STOCs and TICCs.

We investigated the effects of Synta66, a more selective and potent Orai1 inhibitor compared with BTP2, on STOCs and TICCs in VSMCs (Figure 7—figure supplements 1 and 2) and myogenic reactivity in isolated cerebral arteries (Figure 8—figure supplement 1). We did not observe any effect of Synta66 on STOCs, TICCs, or vasoconstriction, indicating that these are independent of Orai1 activity.

5. Another possibility is that STIM1 influences the accumulation of channels at SR-PM junctions. Do clusters of STIM1 colocalize with clusters of BK and TRPM4? Does STIM1 KO affect the number of channels at junctions? This will contribute to the size of the local STOC and TICC signals.

Using the TIRF modality of our GSDIM system, we found that STIM1 colocalized with both BK and TRPM4 at or near the plasma membrane (Figures 5). We did not observe any differences in BK or TRPM4 channel density in VSMCs from *Stim1-*smKO mice compared to controls (Figures 3B and 4B). In addition, we found that STIM1 knockout did not alter whole-cell BK and TRPM4 current density (Figure 7E and I), providing evidence that STIM1 deficit does not alter the amount of channel protein available for activation at the plasma membrane. We also show that STIM1 knockout did not alter mRNA levels of BKα and β subunits (Figure 7—figure supplement 1) or TRPM4 (Figure 7—figure supplement 2).

6. It is unclear what proportion of the detected channel clusters are actually located at junctions. GSDIM was performed in epifluorescence mode, which implies that signals were collected throughout the cell thickness, not necessarily near the PM. Cluster densities for the channels differ: BK (15/µm^2^), RyR2 (4-6/µm^2^), TRPM4 (40/µm^2^), and IP3R clusters (30/µm^2^) and are much higher than the density of colocalized clusters (BK-RyR2 are 0.3/µm^2^, and IP3R-TRPM4 are 3/µm^2^). It seems likely that most of these clusters are actually not associated with junctions. If the clusters are not at junctions, then the quantification of cluster density cannot be related to the frequency of STOCs or TICCs.

Our data show that knockout of STIM1 does not alter the density of BK or TRPM4 channel clusters (Figure 3B and 4B) and we make no strong claim that changes in cluster density affect STOC or TICC currents. Instead, we claim that STOC and TICC activity is impaired by knockout of STIM1 because the microdomains that encompass the Ca^2+^ signals necessary for the activation of these channels are compromised. Our data (Figure 2) showing that interactions between the SR and PM are disrupted support this conclusion.

References:1. Sato D, Hernandez-Hernandez G, Matsumoto C, Tajada S, Moreno CM, Dixon RE, et al. A stochastic model of ion channel cluster formation in the plasma membrane. J Gen Physiol. 2019;151(9):1116-34.2. Gonzales AL, Garcia ZI, Amberg GC, Earley S. Pharmacological inhibition of TRPM4 hyperpolarizes vascular smooth muscle. Am J Physiol Cell Physiol. 2010;299(5):C1195-C202.3. Bautista DM, Lewis RS. Modulation of plasma membrane calcium-ATPase activity by local calcium microdomains near CRAC channels in human T cells. J Physiol. 2004;556(Pt 3):805-17.4. Blaustein MP, Lederer WJ. Sodium/calcium exchange: its physiological implications. Physiological reviews. 1999;79(3):763-854.5. Shmigol AV, Eisner DA, Wray S. The role of the sarcoplasmic reticulum as a Ca^2+^ sink in rat uterine smooth muscle cells. J Physiol. 1999;520 Pt 1(Pt 1):153-63.6. Fernandez RA, Wan J, Song S, Smith KA, Gu Y, Tauseef M, et al. Upregulated expression of STIM2, TRPC6, and Orai2 contributes to the transition of pulmonary arterial smooth muscle cells from a contractile to proliferative phenotype. Am J Physiol Cell Physiol. 2015;308(8):C581-93.7. Potier M, Gonzalez JC, Motiani RK, Abdullaev IF, Bisaillon JM, Singer HA, et al. Evidence for STIM1- and Orai1-dependent store-operated calcium influx through ICRAC in vascular smooth muscle cells: role in proliferation and migration. Faseb j. 2009;23(8):2425-37.8. Jousset H, Frieden M, Demaurex N. STIM1 knockdown reveals that store-operated Ca^2+^ channels located close to sarco/endoplasmic Ca^2+^ ATPases (SERCA) pumps silently refill the endoplasmic reticulum. J Biol Chem. 2007;282(15):11456-64.9. Emrich SM, Yoast RE, Xin P, Arige V, Wagner LE, Hempel N, et al. Omnitemporal choreographies of all five STIM/Orai and IP(3)Rs underlie the complexity of mammalian Ca(2+) signaling. Cell reports. 2021;34(9):108760.10. Zheng S, Zhou L, Ma G, Zhang T, Liu J, Li J, et al. Calcium store refilling and STIM activation in STIM- and Orai-deficient cell lines. Pflugers Archiv : European journal of physiology. 2018;470(10):1555-67.11. Feil S, Valtcheva N, Feil R. Inducible Cre mice. Methods in molecular biology (Clifton, NJ). 2009;530:343-63.12. Herring BP, Hoggatt AM, Burlak C, Offermanns S. Previously differentiated medial vascular smooth muscle cells contribute to neointima formation following vascular injury. Vascular cell. 2014;6:21.13. Kühbandner S, Brummer S, Metzger D, Chambon P, Hofmann F, Feil R. Temporally controlled somatic mutagenesis in smooth muscle. Genesis (New York, NY : 2000). 2000;28(1):15-22.14. Wirth A, Benyó Z, Lukasova M, Leutgeb B, Wettschureck N, Gorbey S, et al. G12-G13-LARG-mediated signaling in vascular smooth muscle is required for salt-induced hypertension. Nat Med. 2008;14(1):64-8.15. Thakur P, Dadsetan S, Fomina AF. Bidirectional coupling between ryanodine receptors and Ca^2+^ release-activated Ca^2+^ (CRAC) channel machinery sustains store-operated Ca^2+^ entry in human T lymphocytes. J Biol Chem. 2012;287(44):37233-44.16. Sampieri A, Santoyo K, Asanov A, Vaca L. Association of the IP3R to STIM1 provides a reduced intraluminal calcium microenvironment, resulting in enhanced store-operated calcium entry. Sci Rep. 2018;8(1):13252.17. Pacheco J, Dominguez L, Bohórquez-Hernández A, Asanov A, Vaca L. A cholesterol-binding domain in STIM1 modulates STIM1-Orai1 physical and functional interactions. Sci Rep. 2016;6:29634.18. Lewis AH, Grandl J. Mechanical sensitivity of Piezo1 ion channels can be tuned by cellular membrane tension. *eLife*. 2015;4.19. Gonzales AL, Yang Y, Sullivan MN, Sanders L, Dabertrand F, Hill-Eubanks DC, et al. A PLCgamma1-dependent, force-sensitive signaling network in the myogenic constriction of cerebral arteries. Sci Signal. 2014;7(327):ra49.20. Pires PW, Ko E-A, Pritchard HAT, Rudokas M, Yamasaki E, Earley S. The angiotensin II receptor type 1b is the primary sensor of intraluminal pressure in cerebral artery smooth muscle cells. J Physiol. 2017;595(14):4735-53.21. Gonzales AL, Amberg GC, Earley S. Ca^2+^ release from the sarcoplasmic reticulum is required for sustained TRPM4 activity in cerebral artery smooth muscle cells. Am J Physiol Cell Physiol. 2010;299(2):C279-C88.22. Gonzales AL, Earley S. Endogenous cytosolic Ca(2+) buffering is necessary for TRPM4 activity in cerebral artery smooth muscle cells. Cell Calcium. 2012;51(1):82-93.23. Lachmanovich E, Shvartsman DE, Malka Y, Botvin C, Henis YI, Weiss AM. Co-localization analysis of complex formation among membrane proteins by computerized fluorescence microscopy: application to immunofluorescence co-patching studies. Journal of microscopy. 2003;212(Pt 2):122-31.

[Editors' note: further revisions were suggested prior to acceptance, as described below.]

The manuscript has been improved but there are some remaining issues that need to be addressed, as outlined below:(1) The manuscript does not actually show that STIM1 is present at the SR-PM junctions that are the focus of the paper. Rather, the data in Figure 2 only shows that there is an overall decrease in the number of SR-PM junctions in the STIM1 KO. In the absence of direct proof that STIM1 is located at the junctions, the idea that STIM1 drives the formation of the SR-PM contacts is speculative. Therefore, the title and the discussion of the paper should reflect this point as indicated by reviewers 1 and 3.

We have modified the manuscript's title and altered our discussion of this topic throughout the manuscript as recommended. (Line 113, line 406-410, line 437, line 524).

(2) This point was previously raised. The fraction of BK (and TRPM4 channels) that are actually co-localized with STIM1 is really quite small to begin with. This is readily apparent in the co-localization images of Figure 5 – most BK and TRPM4 channels don't cluster with STIM1. Authors must explain explicitly the relationship between channel clusters and coupling sites in the text. See also comments by reviewers 1 and 3 on this issue.

We cannot image SR and PM dyes in native SMCs using GSDIM because the high laser levels and long exposure times required for this technique completely bleach the dyes. The SIM mode of our LLS instrument is ideal for the imaging of the dyes due to low bleaching but lacks the resolution of the GSDIM system needed for the detection of protein clusters. Consequently, we cannot simultaneously image the sites of membrane interaction and protein clusters. We acknowledge this technical limitation in the revised manuscript. In addition, we provide an explicit explanation of the relationship between channel clusters and coupling sites based on our data (line 442). We have also clarified how the colocalization data shown in Figure 5 were analyzed (Line 251, line 657).

(3) Figure 2 clearly shows that the volume of the SR is strongly diminished in the STIM1 KO. It seems natural to expect then that as the volume of the SR declines and it pulls away from the membrane, SR-PM contacts will also go down. This issue was raised in the previous review. Does this decrease in SR volume alone not account for the observed decline in SR-PM contacts? This question needs to be explicitly addressed in the manuscript with a well-rounded discussion of what the data actually show.

We are in complete agreement – when STIM1 is knocked out in SMC, the SR volume is reduced because the peripheral SR retracts, diminishing SR-PM coupling. These data indicate that STIM1 is involved in maintaining contact between the peripheral SR and the PM necessary for membrane coupling. We explicitly discuss this issue in the revised manuscript. (Line 181, line 191, line 462-483).

Reviewer #1:Reviewers have addressed my previous comments well and the paper is much improved.I do have two further points that would need addressing:– First, nowhere in the paper is there data showing that STIM1 is actually present at the peripheral ER-PM sites. I agree that the data showing that STIM1 KO decreases interactions between the PM and SR are in VSMCs is strong and convincing. But the paper strongly implies that STIM1 forms ("fosters") these contacts. Can the authors show that STIM1 labelling is present at the SR-PM contact sites? In the absence of this data, the conclusion that STIM1 is crucial for "fostering SR-PM coupling" is indirect and in the worst case, could be entirely incidental.

The proposed experiment is a great idea. However, we cannot image SR and PM dyes in native SMCs using GSDIM because high laser levels and long exposure times required for this technique completely bleach the dyes. The SIM mode of our LLS instrument is ideal for the imaging of the dyes (low bleaching) but lacks the resolution of the GSDIM system needed for the detection of protein clusters. Consequently, we cannot simultaneously image the sites of membrane interaction and protein clusters. This technical limitation is acknowledged in the revised manuscript (Line 442). To further address the reviewer's concerns, we've altered the manuscript's title and discussion in several places throughout the manuscript.

– Figure 5: The co-localization of STIM1 with BK (and TRPM4) is actually pretty hard to see and less than compelling. Arrows should be used on the BK and TRPM4 panels to precisely denote which BK clusters are co-localized with which STIM1 clusters. I also don't understand the statements indicating that "BK and TRPM4 colocalize with STIM1 more frequently than would be predicted if the proteins are randomly distributed". How was the random distribution modelled? How many channels? What is the area of the membrane? How was cluster formation modelled/induced in the random distribution model? With sufficient numbers of BK and STIM1 proteins, one would expect that random distribution may even cause co-localization of the observed numbers of BK and STIM1 proteins. These important details that are crucial to understanding whether co-localization is mechanistically driven are missing.

Arrows have been added to Figure 5 to show examples of colocalized clusters. In addition, A more detailed description of the colocalization analysis methods has been added to the revised manuscript (Line 251, line 657).

Reviewer #3:The authors have done a good job of addressing the comments and the manuscript has been improved. There are only a few remaining concerns that I would like the authors to address.1. As for the question of whether channel clusters are present at SR-PM junctions, the authors should explain more clearly in the text what fraction of the clusters are likely to reside at these sites. See my comment on point #2, below.

Please see our response to similar comments from the editor and reviewer 1.

4. Good – looks like there really isn't any contribution from SOCE. In Figure 1—figure supplement 3E, please state how the comparative SOCE response was measured – peak 340/380 ratio, or the slopes from the data in D?

The data reported in Figure 1 —figure supplement 3E are the peak F_340_/F_380_ ratio. We have added this information to the revised manuscript. (Methods, line 580; Figure 1—figure supplement 3 legend; line 1163).

My review:1. My comment on the title did not dispute the effect of STIM1 KO on the number of peripheral coupling sites. The problem is that the wording suggests that STIM1 plays a direct role in forming these sites, and that is still to my knowledge an unanswered question in the field. Elsewhere in the paper the authors state that STIM1 maintains coupling, or is important for stable coupling, which is accurate. However, the title implies STIM1 acts in the formation of junctions, and without more mechanistic studies this is still an open question. I would suggest rewording the title to make it more accurate – perhaps referring to these sites as STIM1-dependent rather than STIM1-formed.

We have modified the title as suggested.

2. My comment relates to the lack of correspondence between channel clusters and coupling sites. As I stated, STIM1 KO greatly reduces coupling site density, but does not change BK, TRPM4 or IP3R cluster density. In addition, coupling sites are 0.17/µm2 in WT, but clusters are much more prevalent: 16/µm2 for BK, 40/µm2 for TRPM4, 32 for IP3R. This 100x difference indicates that many of the channel clusters are not associated with an identified coupling site, and probably for this reason the number is not grossly affected by loss of coupling sites. This conclusion is supported by the data of Figure 5, where only a small fraction of BK and TRPM4 channels colocalize with STIM1 puncta (indicating SR-PM junctions). It would be helpful to explain explicitly the relationship between channel clusters and coupling sites in the text. I may be missing something, but I do not understand how stochastic self-assembly in the rebuttal explains how loss of STIM1 affects cluster size. It is hard for me to imagine that given the very small fraction of channel clusters associated with SR-PM junctions (~5%) in WT cells, how the observed reduction in SR-PM coupling sites could lead to a measurable cell-wide decrease in the size of clusters as the authors suggest.

We agree with your comment – our data indicate that channel clusters are uniformly distributed throughout the membrane and are not enriched at coupling sites. We explicitly discuss this in the revised manuscript (Line 442).

As for how the loss of STIM1 affects protein cluster size- superresolution imaging data from many laboratories demonstrates that all (or almost all) proteins form clusters in membrane with an exponential cluster size distribution. Stochastic self-assembly is a currently accepted theoretical model that accounts for this, and it is backed by rigorous experimental evidence. When asked by the reviewers to explain why we unexpectedly saw a reduction in protein cluster sizes in STIM1 knockout mice, we utilized this model to interpret our data. We agree that loss of coupling sites alone cannot explain the effects on channel clusters distributed throughout the membrane and not present in coupling sites. However, our data show that the entire peripheral SR is no longer associated with the PM following STIM1 knockout. Based on these observations, we put forth the concept that the peripheral SR prolongs the dwell time of channel proteins in the membrane, allowing them to grow to a larger size. STIM1 knockout detaches the peripheral SR from PM, removing its protection against recycling pathways. This effect decreases membrane dwell time and reduces cluster size. The revised manuscript explicitly discusses the potential impact of the peripheral SR in membrane protein cluster size regulation (Lines 462-483).